# A novel perivascular cell population in the zebrafish brain

Marina Venero Galanternik[1], Daniel Castranova[1], Aniket V Gore[1], Nathan H Blewett[1], Hyun Min Jung[1], Amber N Stratman[1], Martha R Kirby[2], James Iben[1], Mayumi F Miller[1], Koichi Kawakami[3,4], Richard J Maraia[1], Brant M Weinstein[1]*

[1]Division of Developmental Biology, Eunice Kennedy Shriver National Institute of Child Health and Human Development, National Institutes of Health, Bethesda, United States; [2]Translational and Functional Genomics Branch, National Human Genome Research Institute, National Institutes of Health, Bethesda, United States; [3]Division of Molecular and Developmental Biology, National Institute of Genetics, Mishima, Japan; [4]Department of Genetics, SOKENDAI (The Graduate University for Advanced Studies), Mishima, Japan

**Abstract** The blood-brain barrier is essential for the proper homeostasis and function of the CNS, but its mechanism of function is poorly understood. Perivascular cells surrounding brain blood vessels are thought to be important for blood-brain barrier establishment, but their roles are not well defined. Here, we describe a novel perivascular cell population closely associated with blood vessels on the zebrafish brain. Based on similarities in their morphology, location, and scavenger behavior, these cells appear to be the zebrafish equivalent of cells variably characterized as Fluorescent Granular Perithelial cells (FGPs), perivascular macrophages, or 'Mato Cells' in mammals. Despite their macrophage-like morphology and perivascular location, zebrafish FGPs appear molecularly most similar to lymphatic endothelium, and our imaging studies suggest that these cells emerge by differentiation from endothelium of the optic choroidal vascular plexus. Our findings provide the first report of a perivascular cell population in the brain derived from vascular endothelium.

*For correspondence: flyingfish2@nih.gov

Competing interests: The authors declare that no competing interests exist.

## Introduction

Homeostatic balance in an organism relies on the adequate function of specialized organs and systems including the blood and lymphatic vasculature. The blood vascular (circulatory) system transports nutrients, oxygen, hormonal signals, and immune cells throughout the body. The lymphatic system is critical for fluid homeostasis, lipid absorption, and reincorporation of macromolecules extravasated into the interstitial spaces of tissues back into the circulation. Defects in the development or function of the lymphatic system can lead to embryonic death, or to debilitating lymphedema in children and adults. Lymphedema results from a failure to drain interstitial fluid via the lymphatics, leading to severe edema of the tissues, and is one of the most common secondary effects of cancer-related prophylactic surgeries and treatments (*Armer and Stewart, 2005*; *Poage et al., 2008*). Molecular characterization has led to the identification of genes essential for lymphatic development. The Prox1 gene is regarded as a lymphatic 'master regulator' based on a series of studies performed in mouse, zebrafish, *Xenopus* and chick (*Hong et al., 2002*; *Oliver and Harvey, 2002*; *Peyrot et al., 2010*; *Rodriguez-Niedenführ et al., 2001*; *Wigle et al., 2002*; *Wigle and Oliver, 1999*; *Yaniv et al., 2006*). Work in the zebrafish and other models has highlighted important roles for *lyve1*, *ccbe1*, and other genes in lymphatic development

(*Hogan et al., 2009*; *Okuda et al., 2012*). Although it had been accepted that lymphatic vessels and related organs are found in every tissue of the body with the exclusive exception of the bone marrow and the CNS, a recent study showed that mouse dural sinuses host functional lymphatic vessels capable of performing conventional lymphatic functions (*Louveau et al., 2015*). These CNS-associated lymphatic vessels express classic lymphatic markers such as Lyve1 and Prox1 but it is unclear whether these vessels actually form from preexisting veins like lymphatic vessels elsewhere in the body (*Louveau et al., 2015*). Prior to the discovery of lymphatics in the murine CNS, the lymphatic system was not thought to be involved in regulation of fluid homeostasis or maintenance of blood brain barrier (BBB) integrity in the brain.

Under homeostatic conditions, the BBB is responsible for protecting the brain from the entry of pathogens, neurotoxic molecules and lipophilic elements (*Ballabh et al., 2004*). CNS-associated endothelial cells and a variety of specialized CNS perivascular cells including astrocytes, pericytes, microglia, and perivascular macrophages (PVMs) are thought to be important for BBB function in the brain (*Williams et al., 2001*). While a great deal is known about the role of pericytes and microglia in the BBB, relatively little is known about the role of PVMs in BBB formation and function. PVMs have been reasonably well described anatomically and histologically (*Mato et al., 1986a*; *Mato and Mato, 1983*, *1979*; *Mato et al., 1981*, *1985*, *1986b*, *1997*, *1998*, *2002*), yet their biological function remains unclear. Based on their known scavenging properties, PVMs appear to be important for protecting the brain from toxic or potentially damaging elements such as lipids, heavy metals, and complex sugars, among others (*Linehan et al., 2000*; *Mato et al., 1997*, *1982b*, *1984*, *2002*; *Mendes-Jorge et al., 2009*). Interestingly, recent reports point to an important role for PVMs in brain vascular permeability regulation and metabolic function (*He et al., 2016*; *Jais et al., 2016*). Clodronate-dependent ablation of PVMs in the mouse showed that at base line, PVMs help suppress vascular permeability and maintain the integrity of the blood vascular barrier, and when VEGF was added in combination with PVM ablation, vessel leakage increased significantly (*He et al., 2016*). On the metabolic side, another recent study showed that the saturated fatty acids found on high fat diets reduce Glucose Receptor 1 (GLUT1) expression in the BBB of mice, impeding brain glucose uptake, but this life threatening effect is ameliorated by the production of VEGF by myeloid-derived PVMs, driving the upregulation of GLUT1 back to homeostatic levels (*Jais et al., 2016*).

Brain perivascular cells called 'Fluorescent Granular Perithelial Cells' (FGPs) were first identified as a result of their yellow fluorescence due to the accumulation of intracellular vesicles containing autofluorescent lipid breakdown products (*Mato et al., 1981*). FGPs are found in the leptomeningeal layers and cerebral cortex of mammals, where they are thought to provide an important pinocytotic protective function, leading to encapsulation of particles within the cells and giving them their stereotypical 'honeycomb like' morphology in light and electron micrographs (*Mato et al., 1986a*, *2002*). The internal vesicles present in FGPs increase with age, and studies have correlated increased FGP vesicle accumulation with onset of cognitive neurological impairment in conditions such as Alzheimer's disease and in lipid metabolic disorders (*Mato and Ookawara, 1981*; *Mato et al., 1981*, *1982a*). The difference between PVMs and FGPs and the exact relationship between them remains unclear at present, including whether these represent different cell types, or if FGPs are in fact the same as or a sub-type of PVMs.

Several studies in mammals have suggested that PVMs and/or FGPs are bone marrow-derived cells (*Audoy-Rémus et al., 2008*; *Faraco et al., 2016*). Specifically, PVMs are thought to form when mature monocytes extravasate from brain vessels in response to pro-inflammatory cytokines (*Audoy-Rémus et al., 2008*). The fact that PVMs strongly express CD206 (aka Mannose Receptor 1, MRC1), a receptor commonly expressed by tissue resident macrophages required to mediate endocytosis of glycoproteins (*Apostolopoulos and McKenzie, 2001*; *Faraco et al., 2016*; *Linehan et al., 2000*), has also been taken as an indication of a macrophage identity for these cells. As noted above, CD206(+) PVM levels increase during a high fat diet and PVMs restore the balance of brain glucose uptake by secreting VEGFA. VEGF myeloid-specific deletion (using a Lys:Cre mouse) produced by CD206-positive cells during high fat diet feeding leads to failure in glucose uptake, consistent with a hematopoietic origin for these cells (*Jais et al., 2016*). Mannose Receptors recognize complex sugars on the surface of pathogens, facilitating phagocytic engulfment and neutralization of viruses, bacteria and fungi (*Apostolopoulos and McKenzie, 2001*; *Gazi and Martinez-Pomares, 2009*; *Linehan et al., 2000*; *Op den Brouw et al., 2009*; *Sweet et al., 2010*). A recent study in which

murine ear skin was inoculated with *Staphylococcus aureus* also showed that PVMs play an important role in recruiting neutrophils to the infection/injury site by secreting a battery of chemokines (*Abtin et al., 2014*). As more studies on PVMs emerge the term FGPs has become less often used. However, it is interesting to note that while PVMs have been described in the brain cerebral cortex, as well as in the retina, skin, peritoneum, mesentery and cremaster muscle (*Abtin et al., 2014*; *He et al., 2016*; *Jais et al., 2016*; *Mendes-Jorge et al., 2009*), FGPs with their highly characteristic autofluorescent vesicles have only been described in the mammalian brain and in the retina (*Mato et al., 1981*; *Mendes-Jorge et al., 2009*), suggesting that FGPs may be different from or define a subset of PVM cells.

Here, we identify a new endothelial-derived cell population in the zebrafish brain analogous to murine FGPs. Taking advantage of the live imaging capabilities of the zebrafish, we show that zebrafish FGPs are not derived from primitive or definitive hematopoietic progenitors, but instead emerge early in development from the endothelium of the optic choroidal vascular plexus, a primitive endothelial vascular plexus that resides deep within the brain behind the eyes, from where FGPs detach and migrate to populate blood vessels over the surface of much of the brain. Zebrafish FGPs do not express markers of other well-characterized perivascular cell types, but they do express lymphatic endothelial markers. However, zebrafish FGPs do not form vessels, but remain as individual cells in close association with brain blood vessels. This work represents the first report of FGPs in the zebrafish CNS, providing a powerful new model for future *in vivo* studies characterizing the role of these novel and unusual cells in health and disease.

## Results

### Mrc1a-positive perivascular cells cover the zebrafish brain

In light of recent reports describing lymphatic vessels in the mammalian brain (*Louveau et al., 2015*), we decided to more closely examine the CNS vasculature of the zebrafish. We imaged brains dissected from double-transgenic adult zebrafish carrying both a *Tg(mrc1a:eGFP)* transgene marking primitive veins and lymphatic vessels, and a *Tg(kdrl:mCherry)* transgene marking blood vessels (*Jung et al., 2017*). Although we observe some Mrc1a:eGFP-positive lymphatic vessels around the brain (see below) we also found very large numbers of individual, strongly Mrc1a:eGFP-positive cells covering the surface of much of the adult brain (2551.25 ± 248.35 S.D. cells/dorsal side of brain, n = 4, *Figure 1a,b*, *Figure 1—figure supplement 1*). These cells are particularly dense on the surface of the optic tectum (*Figure 1a,b*, *Figure 1—figure supplement 1a*), but they are also found on the dorsal side of the cerebellum, the ventral side of the hypothalamus and on the olfactory bulbs (*Figure 1—figure supplement 1b*). Higher magnification confocal imaging shows that these strongly GFP-positive cells are mostly disconnected cells that do not form tubes, although they are found very closely apposed to blood vessels (*Figure 1c,d*). High-resolution confocal imaging of individual perivascular cells shows they are flat, elongated spindle-shaped cells 40 μm (39.46 ± 11.61 S.D., n = 218) in length. Cross sections of the adult zebrafish brain confirmed that these perivascular cells reside only in the most exterior meningeal layers (*Figure 1e–g*).

Mrc1a:eGFP-positive perivascular cells are also present on the surface of the brain in developing zebrafish, as can be seen in 10, 15, and 30 dpf animals (*Figure 2a–g*). Despite having extensive processes often extending far along adjacent blood vessels (*Figure 2f,g*), these cells appear to be relatively stationary at all but early developmental stages (see below). The Mrc1a-positive perivascular cells on the surface of the tectum increase in number over time but appear EdU-negative during development, suggesting they are not proliferating there but arriving from elsewhere (*Figure 2—figure supplement 1a*). Interestingly, Mrc1a:eGFP-positive brain perivascular cells are also marked by several other transgenes known to be expressed in lymphatic or lympho-venous endothelial cells, including Lyve1:dsRed (*Figure 2h–j*) and Prox1aBAC:KalTA4-4xUAS-E1b:uncTagRFP (from here on called Prox1a:RFP) (*Figure 2k–m*), suggesting a potential lymphatic lineage relationship. However, Mrc1a-GFP-positive cells are not part of either blood or lymphatic vessels, do not form tubes and, in contrast to other lymphatic vessels surrounding the brain, Mrc1a-GFP-positive cells do not take up intramuscularly injected Q-dots (*Figure 2—figure supplement 1b,c*).

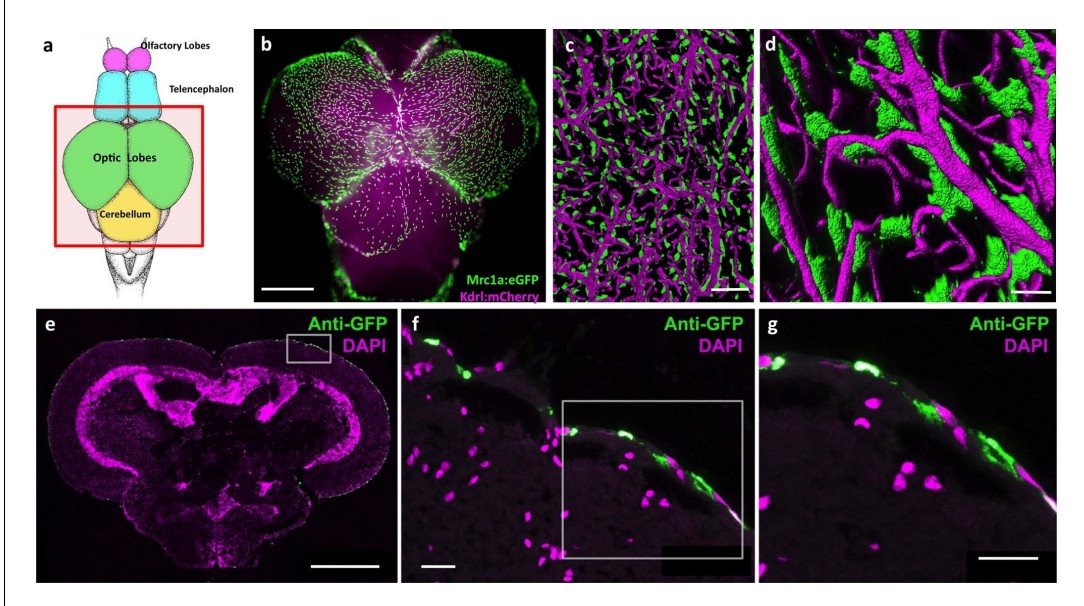

**Figure 1.** Mrc1a-positive perivascular cells cover the zebrafish brain. (a) Schematic diagram of an adult zebrafish brain. Dorsal view, rostral at top. Box shows region imaged in panel b. (b) Epifluorescence microscopic image of the optic lobes (top) and cerebellum of the dissected brain of a *Tg(mrc1a: eGFP);Tg(kdrl:mCherry)* double-transgenic 4 month old adult zebrafish (eGFP and mCherry are shown in green and magenta, respectively). (c,d) Higher magnification confocal images of Kdrl:mCherry-positive blood vessels (magenta) and closely associated Mrc1a:eGFP-positive perivascular cells (green) on the surface of one of the optic lobes of the brain. (e) Confocal image of a transverse section through the brain of a *Tg(mrc1a:eGFP)* transgenic adult zebrafish, at the level of the optic lobes (top), stained with anti-GFP (green) and DAPI (magenta), n = 2 brains. Box shows region depicted in panel F. (f, g) Higher magnification confocal images of the brain section imaged in panel E, showing Mrc1a:eGFP-positive cells restricted almost exclusively to the surface meningeal layer. Box in f shows region imaged in panel g. Scale bars: 500 μm (b,e), 100 μm (c), 50 μm (d), 20 μm (f,g).

The following figure supplement is available for figure 1:

**Figure supplement 1.** Mrc1a-positive perivascular cells cover the zebrafish brain.

## Mrc1a-positive perivascular cells are fluorescent granular perithelial cells (FGPs)

Closer observation revealed that Mrc1a:eGFP-positive perivascular cells strongly resemble a novel population of mammalian cells variably described as Fluorescent Granular Perithelial cells (FGPs), Perivascular Macrophages (PVMs), or 'Mato Cells.' These mammalian cells were described by Masao Mato in 1979 (*Mato and Ookawara, 1979*) as a specialized form of PVM closely associated with blood vessels on the surface of the mammalian brain cortex. The most unique and distinguishing characteristic of mammalian FGPs is the presence of numerous yellow autofluorescent 'granules' identified as large intracellular vesicular/vacuolar compartments filled with weakly autofluorescent lipid breakdown products, in addition to other materials (*Mato et al., 1981*). Fluorescent imaging of dissected brains from wildtype, non-transgenic adult zebrafish reveals that these cells are in close proximity to blood vessels on the cortical surface of the brain (*Figure 3a,b*). Higher magnification confocal imaging of perivascular cells on the brains of *Tg(mrc1a:eGFP)* or *Tg(lyve1:dsRed)* adult zebrafish reveals that they contain numerous large intracellular vesicular/vacuolar compartments that emit weak yellow autofluorescence (*Figure 3c–f*). As in the zebrafish, mammalian FGPs are found in the superficial leptomeningeal layers of the brain (*Mato and Ookawara, 1979*; *Mato et al., 1981*, *1985*), and we find that murine FGPs are also strongly positive for MRC1 (CD206) (*Figure 3g,h*).

Another distinguishing feature of mammalian FGPs is their capacity to 'scavenge' extracellular macromolecules and heavy molecular weight particles including India ink, horseradish peroxidase and ferritin, taking them up and incorporating them into their intracellular vacuoles (*Mato et al., 1984*). India Ink injected into the brains of adult *Tg(mrc1a:eGFP)* zebrafish is taken up by Mrc1a: eGFP-positive FGPs, as evidenced by the appearance of dark vesicles (*Figure 3i,j*). In mammals

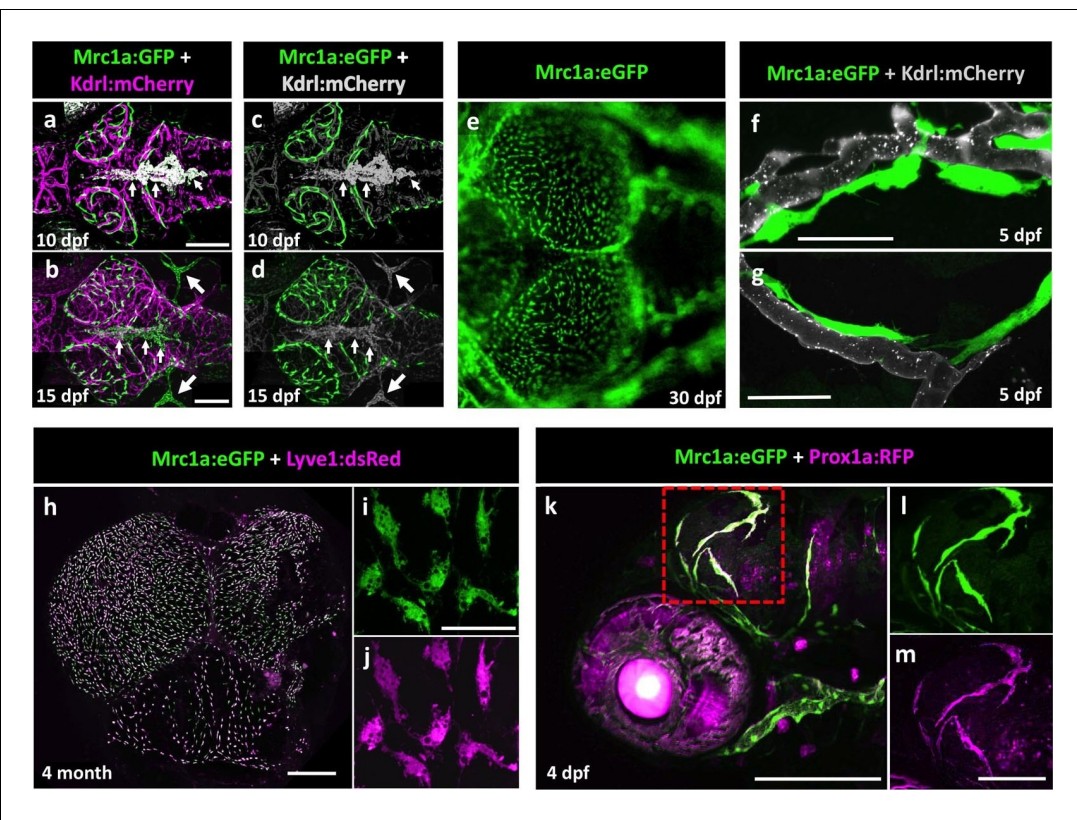

**Figure 2.** Mrc1a-positive perivascular cells are present in the developing zebrafish brain, and express lymphatic markers. (a,b) Confocal images of Mrc1a:eGFP-positive cells (green) and Kdrl:mCherry-positive blood vessels (magenta) on the surface of the brain in 10 dpf (a) and 15 dpf (b) *Tg(mrc1a:eGFP);Tg(kdrl:mCherry)* double-transgenic zebrafish (n = 3 animals of each age imaged). (c,d) Same images as in panels (a) and (b) with both Kdrl:mCherry blood vessel fluorescence and non-perivascular green fluorescence rendered in grey, Mrc1a-positive perivascular cells are depicted in green. Small arrows in panels a-d are autofluorescent pigment cells on the dorsal head of the zebrafish. Large arrows in panels b and d are facial lymphatics. (e) Epifluorescence microscopic image of Mrc1a:eGFP-positive cells on the optic lobes (left) of the dissected brain of a 1 month-old *Tg(mrc1a:eGFP)* transgenic zebrafish (n = 3 brains imaged). Panels a-e are dorsal views of the brain surface, with rostral to the left. (f,g) Airyscan confocal images of Mrc1a:eGFP-positive perivascular cells (green) extending along Kdrl:mCherry-positive blood vessels (grey) in 5 dpf *Tg(mrc1a:eGFP);Tg(kdrl:mCherry-CAAX)* double-transgenic zebrafish (n = 3 animals imaged). (h) Confocal image of Mrc1a:eGFP (green), Lyve1:dsRed (magenta) double-positive cells on the dissected brain of an adult 4 month old *Tg(mrc1a:eGFP);Tg(lyve1:dsRed)* double-transgenic zebrafish (dorsal view, rostral up), n = 3 brains imaged. (i,j) Higher magnification single-channel confocal images illustrating that the cells on the surface of the brain imaged in panel h are expressing both Mrc1a:eGFP (i) and Lyve1:dsRed (j). (k) Confocal image of Mrc1a:eGFP (green), Prox1a:RFP (magenta) double-positive cells on 4 dpf brain of a *Tg(mrc1a:eGFP);Tg(prox1:RFP)* double-transgenic animal (lateral view, rostral to the left). l,m, Single channel Mrc1a:eGFP (green; l) and Prox1a:RFP (magenta; m) images of the boxed region in panel k, showing the cells are expressing both transgenes. dpf, days post fertilization. Scale bars: 200 μm (a–d, k), 35 μm (f,g), 500 μm (h), 50 μm (i,j), 100 μm (l,m).

The following figure supplement is available for figure 2:

**Figure supplement 1.** Mrc1a-positive perivascular cells are EdU negative and fail to collect and drain dye like Mrc1a-positive lymphatics.

FGPs are thought to scavenge excess lipids, especially in response to a high fat diet (*Mato et al., 1982a*, *1983*, *1982b*). Similarly, zebrafish FGPs become elongated and accumulate a larger amount of lipid droplets following an acute high cholesterol diet (HCD) (*Figure 3k–n*). Based on the similarities between these cell types, we believe the Mrc1a-positive brain perivascular cells we have identified, are in fact, zebrafish FGPs, and we refer to them as such hereafter.

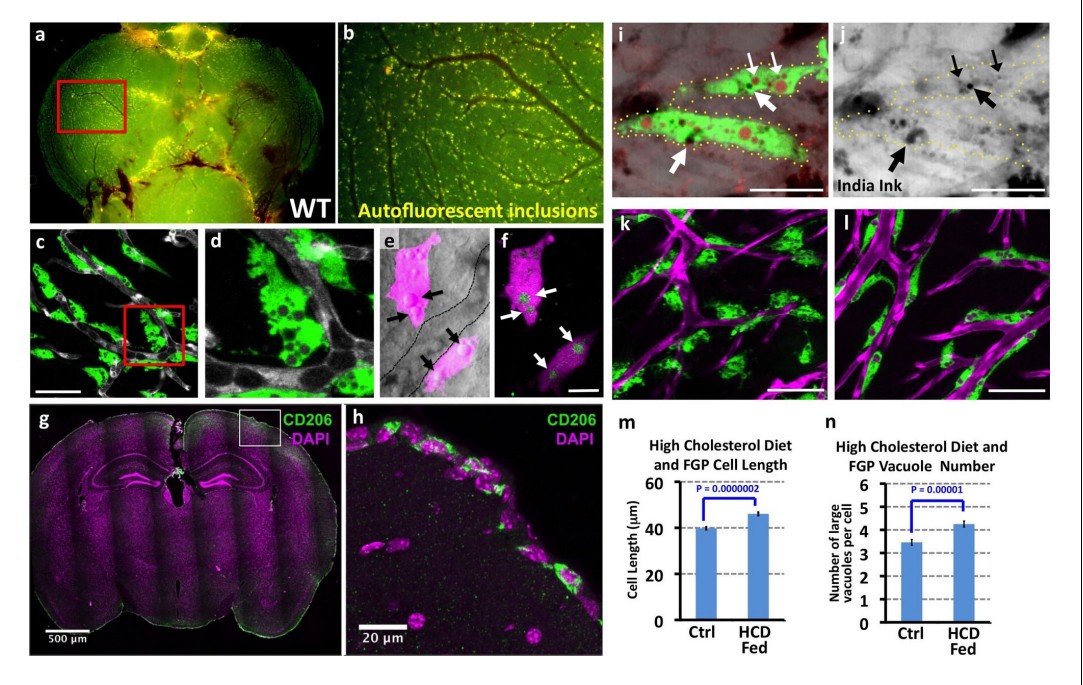

**Figure 3.** Mrc1a-positive perivascular cells are Fluorescent Granular Perithelial cells (FGPs). (a,b) Yellow/green epifluorescence microscopic images of the optic lobes (top) and a portion of the cerebellum of the dissected brain of a non-transgenic wild type adult zebrafish (n = 3 adult brains). Dorsal views rostral at top. Box in a shows region imaged in panel b, where autofluorescent granules are clearly visible in close association with blood vessels. (c,d) Confocal images of Mrc1a:eGFP-positive perivascular cells (green) filled with numerous vacuoles closely apposed to Kdrl:mCherry-positive blood vessels (grey) on the surface of the optic lobe of a dissected brain of a *Tg(mrc1a:eGFP);Tg(kdrl:mCherry)* double-transgenic adult zebrafish (n = 3 adult brains). Box in panel (c) shows region imaged in panel d. (e) Confocal and transmitted light (grey) overlay image of Lyve1:dsRed-positive perivascular cells (magenta) filled with large vacuoles (black arrows) in the brain of an adult *Tg(lyve1:dsRed)* transgenic zebrafish (n = 3 adult brains). Dashed lines demarcate an adjacent blood vessel. (f) Confocal image of the same Lyve1:dsRed-positive cells (magenta) showing that the vacuoles are all yellow autofluorescent (green, white arrows). (g) Confocal image of a transverse section through the brain of an adult mouse, at the level of the cerebral cortex, stained with anti-CD206 (MRC1; green) and DAPI (magenta), n = 3 brains, three sections. (h) Higher magnification confocal images of the brain section imaged in panel g, showing CD206-positive cells on the outermost surface of the brain. (i,j) Confocal images of Mrc1a:eGFP-positive perivascular cells (green) containing numerous vacuoles (red and black) on the surface of an optic lobe of a dissected brain of a *Tg(mrc1a:eGFP)* transgenic adult zebrafish injected intracranially with India ink. Panel (j) shows transmitted light image. Yellow dotted lines in j represent the outline of Mrc1a:eGFP-positive cells. Black vacuoles contain India ink internalized by Mrc1a:eGFP-positive cells (n = 3/10 injected brains). (k,l) Confocal images of Mrc1a:eGFP-positive perivascular cells (green) on Kdrl:mCherry-positive blood vessels (magenta) on the surface of the optic lobes of dissected brain of a normally (k) fed and high cholesterol diet (HCD) fed (l) *Tg(mrc1a:eGFP);Tg(kdrl:mCherry)* double-transgenic adult zebrafish (n = 3 per treatment). (m,n) Quantification of average FGP cell length (panel m; T-test, p-value=$1.7 \times 10^{-10}$, n = 347 FGPs from 3 adult brains) and average vacuole number (panel n; T-test, p-value=$9.2 \times 10^{-6}$, n = 354 FGPs from 3 adult brains) in control versus HCD fed *Tg(mrc1a:eGFP);Tg(kdrl:mCherry)* double-transgenic adult zebrafish. Scale bars: 50 μm (c,d), 10 μm (e,f), 500 μm (g), 20 μm (h), 25 μm (i,j), 50 μm (k,l).

## FGPs do not correspond to other known perivascular cell types

As noted above, zebrafish FGPs are marked by three different transgenes also expressed in lymphatic and/or lympho-venous endothelial cells - Mrc1a, Lyve1, and Prox1a, suggesting a potential relationship with this lineage (*Figure 2f–m*). FGPs reside outside of the vessels but in very close association with them, suggesting they are a type of non-endothelial perivascular supporting cell. A variety of different supporting cell types are known to be present in the perivascular space of brain blood vessels, including pericytes/vascular smooth muscle cells, microglia, radial glia, macrophages and neutrophils (*Figure 4a*). We used transgenes known to mark some of these different perivascular cell lineages to examine whether FGPs might correspond to, or overlap with, one of these previously identified cell types, including *Tg(pdgfrb:Citrine)* for pericytes and/or vascular smooth muscle cells (*Vanhollebeke et al., 2015*), *Tg(mpeg1:gal4);Tg(UAS:NTR-mCherry)* for

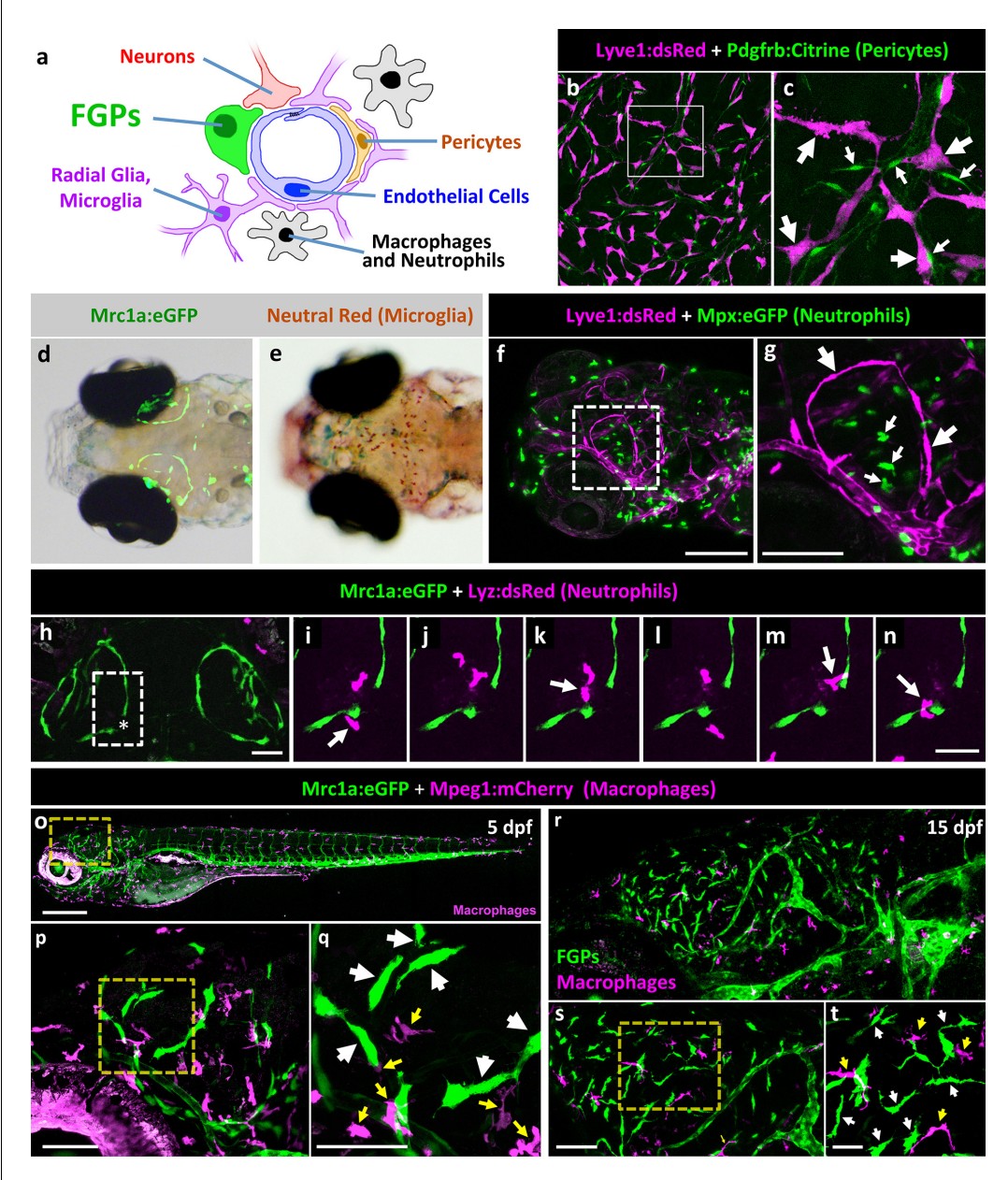

**Figure 4.** Zebrafish FGPs do not correspond to other known perivascular cell types. (a) Schematic diagram of cells associated with brain blood vessels, including the vascular endothelial cells, pericytes/smooth muscle cells, radial glia, macrophages/neutrophils, and FGPs. (b) Confocal image of Lyve1: dsRed-positive (magenta) and Pdgfrb:Citrine-positive (green) cells on the dissected brain of an adult *Tg(lyve1:dsRed);Tg(pdgfrb:Citrine)* double-transgenic zebrafish (n = 2 brains imaged). (c) Higher magnification confocal image of the same field illustrating that the two transgenes are expressed in distinct and separate cell populations - FGPs (large arrows) and pericyte (small arrows). (d) Transmitted light and epifluorescence Mrc1a:eGFP composite image of the dorsal head of a 5 dpf *Tg(mrc1a:eGFP)* transgenic animal (n = 15 animals). (e) Transmitted light image of the dorsal head of a neutral red-stained 5 dpf *Tg(mrc1a:eGFP)* transgenic animal (n = 15 animals). Rostral to the left. (f) Confocal image of Lyve1:dsRed-positive (magenta) and Mpx:eGFP-positive (green) cells in the head of a 5 dpf *Tg(lyve1:dsRed);Tg(mpx:eGFP)* double-transgenic zebrafish (n = 3 animals imaged). Lateral view, rostral to the left. (g) Higher magnification image of the boxed region in panel f, illustrating that the two transgenes are expressed in distinct and separate cell populations - FGP (large arrows) and neutrophils (small arrows). (h–n) Overview image (h) and magnified images of boxed region in panel h at selected time points (i–n) from a time-lapse confocal image series of Lyz:dsRed positive neutrophils (magenta) migrating into and out of the area of a wounded vessel (asterisk in h) on the dorsal brain surface of a 4 dpf *Tg(mrc1a:eGFP);Tg(lyz:dsRed)* double-transgenic zebrafish (rostral up), (n = 2 animals time-lapse imaged). The migrating neutrophils frequently interact with the FGPs adjacent to the wound site (arrows in panels i,k,m,n). (o) Lateral view confocal image of a 5 dpf *Tg(mrc1a:eGFP);Tg(mpeg1:gal4);Tg(UAS:NTR-mCherry)* triple-transgenic zebrafish, with Mrc1a:eGFP-positive cells in green and Mpeg1:Gal4/UAS-NTR-mCherry-positive macrophages in magenta. (p), Higher magnification image of the boxed region in panel o (n = 5

*Figure 4 continued on next page*

*Figure 4 continued*

animals imaged). (**q**) Higher magnification image of the boxed region in panel p, showing that the two transgenes are expressed in distinct and separate FGP (white arrows) and macrophage (yellow arrows) cell populations. (**r**), Lateral view confocal image of the brain (centered on the optic lobes) of a 15 dpf *Tg(mrc1a:eGFP);Tg(mpeg1:gal4);Tg(UAS:NTR-mCherry)* triple-transgenic zebrafish, with Mrc1a:eGFP-positive cells in green and Mpeg1: Gal4/UAS-NTR-mCherry-positive macrophages in magenta (n = 3 animals imaged). (**s**), Higher magnification image of the brain in panel r. Yellow box shows magnified image in panel t. (**t**) Higher magnification image of the boxed region in panel s, showing that the two transgenes are expressed in distinct and separate FGP (white arrows) and macrophage (yellow arrows) cell populations. Rostral is to the left in all images. Scale bars: 100 µm (**b, s, t**), 50 µm (**c, h–n**), 200 µm (**f,g, o–q**).

The following figure supplements are available for figure 4:

**Figure supplement 1.** Radial Glia and FGPs localize to different parts of the brain.

**Figure supplement 2.** FGPs are not Neural Crest derived.

---

macrophages, and *Tg(mpx:eGFP)* (*Renshaw et al., 2006*) and *Tg(lyz:dsRed)* for neutrophils (*Hall et al., 2007*).

Confocal imaging of a *Tg(lyve1:dsRed);Tg(pdgfrb:Citrine)* double-transgenic adult zebrafish shows that although Pdgfrb:Citrine-positive pericytes or vascular smooth muscle cells and Lyve1: DsRed-positive FGPs reside both in close proximity on the surface of the brain, they are separate cell populations (*Figure 4b,c*), with the FGPs representing a population of larger and more elongated cells. In the mammalian cerebral cortex light microscopy and classic counterstains such as PAS-Hematoxylin, and Horseradish Peroxidase show that FGPs and microglia, a specialized population of brain tissue resident macrophages, are separate cell populations (*Mato et al., 1985*). To address the possibility that Mrc1a:eGFP-positive cells are microglia, we used a neutral red vital dye previously shown to strongly label microglial lysosomes in the zebrafish (*Herbomel et al., 2001*). Zebrafish larvae exposed to neutral red showed clear labeling of microglia cells on the dorsal brain at 5 dpf, but these were not on the developing optic tectum where the Mrc1a:eGFP-positive FGPs (which were not stained with neutral red) were located (*Figure 4d,e*). Confocal imaging of *Tg(lyve1: dsRed);Tg(mpx:eGFP)* or *Tg(mrc1a:eGFP);Tg(lyz:dsRed)* double transgenic animals also shows that Lyve1:dsRed-positive FGPs do not express Mpx:eGFP, nor do Mrc1a:eGFP-positive FGPs express Lyz:dsRed, and the FGPs are clearly distinct from Mpx:eGFP or Lyz:DsRed-expressing neutrophils (*Figure 4f–n*). Although few if any Lyz:DsRed-positive neutrophils are normally observed on the brain of 5 dpf developing zebrafish, following laser injury to an FGP-containing brain vessel Lyz:DsRed-positive neutrophils can be observed migrating transiently to the wound area (*Figure 4h–n*). Interestingly, although the Mrc1a:eGFP-positive FGPs adjacent to the wound do not appear to migrate themselves, the migrating neutrophils frequently appear to physically interact with these FGPs (*Figure 4i,k,m,n*). Confocal imaging of 5 dpf or 15 dpf old *Tg(mrc1a:eGFP); Tg(mpeg1:gal4);Tg (UAS:NTR-mCherry)* triple-transgenic zebrafish shows that Mrc1a-eGFP(+) FGPs and Mpeg1-mCherry(+) macrophages are distinct cell populations (*Figure 4o–t*). Finally, we compared the neuroanatomical location of Lyve1-dsRed-positive perivascular cells to the glial cell population labeled with a *Tg(gfap:GFP)* transgenic line and with the GFAP antibody using the Zebrafish Brain Browser software (*Marquart et al., 2015*), and found no overlap between the two cell populations. Glial cells are found throughout deep brain tissues but not on the brain surface, while as noted above, FGPs are located only in the most superficial exterior tissues, which are not Gfap:GFP-positive (*Figure 4— figure supplement 1*). Together, these results support the idea that zebrafish FGPs do not correspond to or overlap with other previously described brain perivascular cell populations in the zebrafish.

## FGPs are not neural crest derived

Neural crest cells are an embryonic transient migratory cell population that populates various target tissues and has a multipotent capacity to differentiate into various cell types, including neurons and glia of the peripheral nervous system, craniofacial skeleton, and pigment cells (*Bhatt et al., 2013*; *Bronner, 2012*; *Bronner and LeDouarin, 2012*; *Knight and Schilling, 2006*). Although our data shows that FGPs are not glial cells, they could still represent a distinct population of neural crest

derived cells. The transcription factor Sox10 is a crucial neural crest specification and maintenance marker expressed specifically by migratory and post migratory neural crest cells as well as by neural crest-derived cells and tissues (*Aoki et al., 2003*; *Carney et al., 2006*; *Dutton et al., 2001*; *Honoré et al., 2003*; *Paratore et al., 2001*). To examine a potential neural crest contribution to FGP formation, we analyzed *Tg(mrc1a:eGFP); Tg(sox10:dsRed)* double transgenics at 3 dpf as FGPs begin to arise and at 5 dpf when they localize to the tectum and become sessile (*Video 1*, *Figure 4—figure supplement 2*). Time lapse imaging shows that 3 dpf emerging FGPs are Sox10:dsRed-negative (*Video 1*, *Figure 4—figure supplement 2a,b*), although Sox10:dsRed-positive pigment cells can be observed migrating on top of the Sox10:dsRed-negative FGPs (*Video 1*). FGPs also lack Sox10 expression at later stages, as shown by imaging FGPs on the 5 dpf optic tectum (*Figure 4—figure supplement 2c–f*). These results show that zebrafish FGPs are not neural crest derived.

## FGPs are not derived from primitive or definitive hematopoietic progenitors

Since mammalian FGPs have been reported to be macrophages, aka 'Perivascular macrophages' (*Mato et al., 1996*; *Williams et al., 2001*), we decided to further investigate the possibility that zebrafish FGPs are derived from the hematopoietic lineage. As described above, we were able to show that FGPs represent a population of cells distinct from Mpx:eGFP-, Lyz:dsRed- and/or Mpeg1:mCherry-positive neutrophils or macrophages. However, the possibility remained that these cells might represent another hematopoietic sub-lineage. During early development, blood cells arise in two distinct waves of hematopoietic differentiation, an earlier primitive wave and a later definitive wave (*Jing and Zon, 2011*). To test whether FGPs are derived from primitive hematopoietic progenitors, we knocked down *pu.1*, a gene required for primitive hematopoiesis (*Rhodes et al., 2005*), in *Tg(mrc1a:eGFP)* embryos. As shown previously, *pu.1* knockdown results in loss of expression of the granulocyte-specific marker *myeloid-specific peroxidase* (*mpx*) above the yolk and the intermediate cell mass (n = 15/15 embryos) at 20 hpf, confirming successful ablation of the primitive hematopoietic wave (*Figure 5—figure supplement 1a–d*). However, FGP numbers at 5 dpf are not significantly different from controls (*Figure 5—figure supplement 1e–i*), confirming that primitive hematopoiesis is not needed for FGP formation.

To assess a potential definitive hematopoietic origin for FGPs, we examined whether FGPs emerge from trunk hematopoietic stem and progenitor cells (HSPCs) like other definitive hematopoietic cell types (*Sood et al., 2010*). During definitive hematopoiesis, HSPCs emerge from endothelial cells on the ventral side of the dorsal aorta beginning at approximately 24–26 hpf (*Boisset et al., 2010*; *Gérard et al., 2010*; *Gore et al., 2016*; *Kissa and Herbomel, 2010*; *Murayama et al., 2006*) (*Figure 5a*). Definitive HSPCs migrate to sites of hematopoiesis such as the thymus and anterior kidney, where they differentiate to give rise to downstream blood lineages (*Kissa and Herbomel, 2010*; *Murayama et al., 2006*). We examined whether HSPCs contribute to brain FGPs by photoconverting Kaede protein in the 24 hpf dorsal aorta endothelium from green to red fluorescence in *Tg(egfl7:gal4);Tg(UAS:kaede);Tg(mrc1a:eGFP)* (*Figure 5b,c*) or *Tg(fli:gal4);Tg(UAS:kaede);Tg(mrc1a:eGFP)* (*Figure 5—figure supplement 2a–c*) triple transgenic animals and then examining whether red fluorescence was subsequently detected in Mrc1a:eGFP-positive brain FGPs. We carefully photoconverted the entire dorsal aorta at 24 hpf and then examined Mrc1a:eGFP-positive FGPs for red fluorescence at 5 dpf. Animals in which the DA was photoconverted at 24 hpf had robust red fluorescence in the 5 dpf dorsal aorta (*Figure 5e* and *Figure 5—figure supplement 2d,e*) and in the thymus (*Figure 5d* and *Figure 5—figure supplement 2f*), but they had no red fluorescent FGPs in the

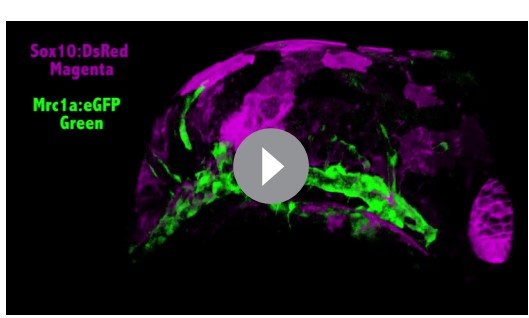

**Video 1.** Confocal time-lapse movie of emerging Fluorescent Granular Perithelial cells in a *Tg(mrc1a:eGFP);Tg(sox10:dsRed)* double transgenic zebrafish embryo from 50 hpf to 70 hpf. *Tg(mrc1a:eGFP)* is displayed in green and *Tg(sox10:dsRed)* is displayed in magenta. Stacks were acquired every 5 min using a 40X water, 1.1NA objective with on a Zeiss LSM880 confocal microscope. Rostral is to the left.

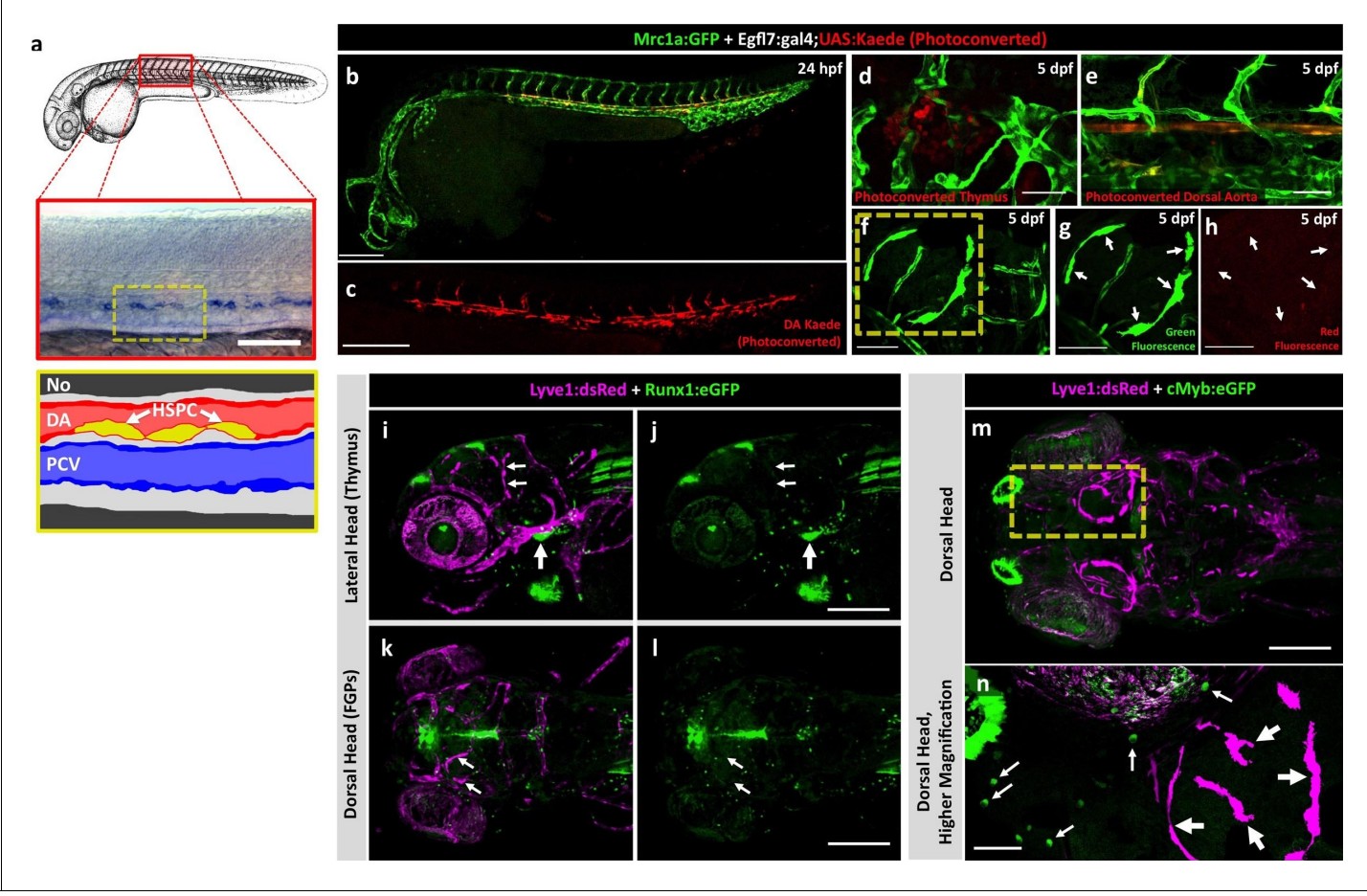

**Figure 5.** Zebrafish FGPs are not derived from definitive hematopoietic progenitors. (a) Schematic diagram illustrating hematopoietic stem and progenitor cells (HSPCs) differentiation from endothelial cells in the ventral floor of the dorsal aorta (DA). No, Notochord; DA, Dorsal aorta; PCV, Posterior Cardinal Vein. (b,c) Green-to-red photoconversion of the DA in a 24 hpf *Tg(egfl7:gal4);Tg(UAS:Kaede);Tg(mrc1a:eGFP)* triple-transgenic embryo, showing red/green (b) or red only (c) confocal fluorescence images immediately after photoconversion at 24 hpf (b,c; n = 5 animals photoconverted). (d-h) Confocal images of green (d–g) and red (d,e,f,h) fluorescence in the thymus (d), trunk vessels (e) and dorsal optic tectum FGPs (f–h) in 5 dpf *Tg(egfl7:gal4);Tg(UAS:Kaede);Tg(mrc1a:eGFP)* triple-transgenic animal subjected to DA photoconversion at 24 hpf. Arrows in panels g and h show Mrc1a:eGFP positive FGPs on the head (g) that are not red fluorescent (h; n = 5/5 photoconverted animals lacking red fluorescent FGPs). Yellow box in notes region shown at higher magnification in panels g and h. (i–l) Confocal imaging of Lyve1:dsRed (magenta; i,k) and Runx1:GFP (green; i–l) fluorescence in the lateral head (i,j) or dorsal head (k,l) of 5 dpf *Tg(lyve1:dsRed);Tg(runx1:eGFP)* double-transgenic animals (rostral to the left in all panels; n = 3 animals imaged). Large arrows in panels i and j note the Runx1:GFP-positive thymus. Small arrows in panels i-l note Lyve1:dsRed-positive but Runx1:GFP-negative FGPs. (m,n) Confocal imaging of Lyve1:dsRed (magenta) and c-Myb:eGFP (green) fluorescence in the dorsal head of a 5 dpf *Tg(lyve1:dsRed);Tg(c-myb:eGFP)* double-transgenic animals (rostral to the left in all panels; n = 3 animals imaged). Boxed region in panel m is displayed at higher magnification in panel n. Large arrows note Lyve1:dsRed-positive but c-Myb:eGFP negative FGPs; small arrows note c-Myb:eGFP-positive but Lyve1:dsRed negative hematopoietic cells. Scale bars: 200 μm (b–c, g–k), 100 μm (d), 50 μm (e,f–h,i–n).

The following figure supplements are available for figure 5:

**Figure supplement 1.** FGPs are not derived from primitive hematopoiesis.

**Figure supplement 2.** Zebrafish FGPs are not derived from definitive hematopoietic progenitors.

**Figure supplement 3.** Inhibiting HSPCs specification does not affect FGP formation.

5 dpf brain (*Figure 5f–h* and *Figure 5—figure supplement 2g,h*; n(*egfl7:gal4*)=5/5; n(*fli:gal4*)=8/8;).

To further rule out a potential hematopoietic ontogeny for zebrafish FGPs, we crossed the *Tg (lyve1:dsRed)* line to two different HSPC reporter lines, *Tg(runx1:eGFP)* and *Tg(c-myb:eGFP)* (*North et al., 2007*), and imaged the resulting *Tg(lyve1:dsRed);Tg(runx1:eGFP)* or *Tg(lyve1:dsRed); Tg(c-myb:eGFP)* double-transgenic animals at 5 dpf (*Figure 5i–n*). No Runx1:eGFP or c-Myb:eGFP expression was detected in Lyve1:DsRed-positive FGPs despite robust GFP expression in the thymus and/or in circulating GFP-positive monocytes (*Figure 5i–n*). Finally, we decided to block the specification of HSPCs by knocking down *runx1* (*Gore et al., 2016*) in *Tg(lyve1:dsRed);Tg(mpx:eGFP)* double transgenic embryos and examining them for the presence of FGPs. Inhibition of HSPC specification by *runx1* knockdown led to a significant downregulation of Mpx-positive cells (*Figure 5—figure supplement 3a–b',e*; *t*-test, p-value=0.012), however, consistent with our previous observations, Lyve1-positive FGP formation and migration to the 5 dpf optic tectum was not affected (*Figure 5—figure supplement 3c–d,f, t*-test, p-value=0.22). Together, these results suggest that embryonic zebrafish FGPs are not derived from primitive or definitive HSPC.

## FGPs express lympho-venous endothelial cell markers

Our results suggest that zebrafish FGPs are not derived from hematopoietic progenitors, but instead express transgenic markers associated with lymphatic or lympho-venous lineage cells. To look more comprehensively at the gene expression profile of FGPs, we performed RNA-seq analysis on FGPs isolated from the brains of adult *Tg(mrc1a:eGFP);Tg(kdrl:mCherry)* double transgenic zebrafish by FACS sorting (*Figure 6a,b*), comparing the gene expression profile of FGPs to that of the entire adult fish (*Figure 6c–f*) or to co-sorted mCherry(+) endothelial cells (*Figure 6—figure supplement 1*). In comparison to the whole fish, FGPs express high levels of lympho-venous endothelial markers such as *mrc1a, lyve1b, stab1, stab2* and *prox1a,* as well as the lymphangiogenic ligand *vegfd* (*Figure 6c*). The same lymphatic markers are also elevated in FGPs when compared to FACS-sorted mCherry-positive blood endothelial cells (*Figure 6—figure supplement 1a*). However, FGPs show significantly lower expression of some key blood endothelial markers, including the *pecam1* endothelial cell-cell adhesion protein, and the pro-angiogenic ligand *vegfaa*, compared to either the whole fish (*Figure 6c*), or especially mCherry-positive blood endothelial cells, where the zebrafish *vegfr2* receptor *kdrl* expression is also significantly low (*Figure 6—figure supplement 1a*). Comparison between FGPs and whole fish RNA-seq profiles also confirmed that FGPs express reduced levels of macrophage markers (*Figure 6d*), astrocyte/glial markers (*Figure 6e*), neurotrophic factors (*Figure 6e*), and pericyte markers (*Figure 6f*). Many of these genes are expressed at similarly low levels in the FACS-sorted mCherry-positive blood endothelial cells (*Figure 6—figure supplement 1b–d*), although the expression levels of some of these genes are even lower in FGPs than in blood endothelial cells (notably, a number of macrophage and pericyte markers). Interestingly, FGPs express higher levels of some pro-angiogenic ligands than blood endothelial cells, including *vegfab, egfl7,* and the receptors *tie1* and *flt4 (vefgr3)* (*Figure 6—figure supplement 1a*). Together with our previous results, global characterization of FGP gene expression strongly suggests that these cells are most closely related to lymphatic or lympho-venous endothelial cells, and that their gene expression is not consistent with macrophage or other hematopoietic-related cell identity.

## FGPs emerge from the endothelium of the optic choroidal vascular plexus

To determine the origins of FGPs, we examined the earliest stages of emergence of these cells during development using *Tg(mrc1a:eGFP);Tg(kdrl:mCherry)* double-transgenic animals. At 3 dpf FGPs are present in low numbers in more lateral positions in the head (*Figure 7a–c*, *Figure 7—figure supplement 1a,b*), but by 4 dpf these cells are present in larger numbers in more dorsal and medial parts of the brain, especially the surface of the optic tectum (*Figure 7d,e*, *Figure 7—figure supplement 1c,d*), suggesting the cells may migrate dorso-medially to the tectum. Confocal time-lapse imaging of *Tg(mrc1a:eGFP); Tg(kdrl:mCherry)* larvae confirmed that FGP progenitors emerge from deep lateral regions on either side of the head behind the eye, migrating dorsally and then medially to the optic tectum along newly-formed blood vessels (*Figure 7f–j, Video 2*). Some FGPs also migrate toward the ventral side of the hindbrain (data not shown). Migratory FGPs are polarized and possess highly active protrusions (*Figure 7f–j,*). Interestingly, once FGPs reach the optic tectum,

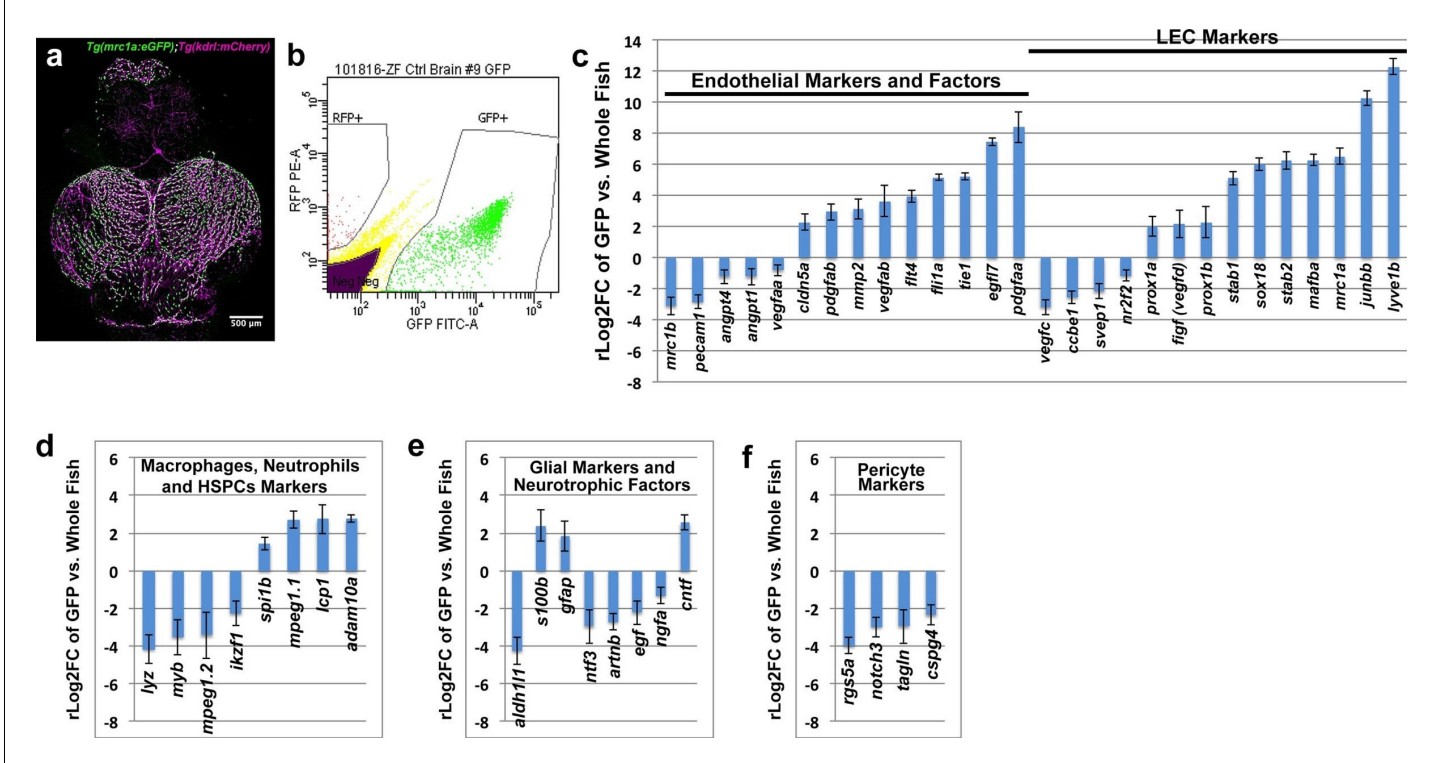

**Figure 6.** Global analysis of gene expression in FACS-sorted adult zebrafish FGPs. (a) Confocal microscopic image of the optic lobes (top) and cerebellum of the dissected brain of a *Tg(mrc1a:eGFP);Tg(kdrl:mCherry)* double-transgenic adult zebrafish (EGFP and mCherry are shown in green and magenta, respectively), n = 10 adult brains. (b) FACS sorting of FGPs (GFP positive) and blood endothelial cells (RFP positive) from *Tg(mrc1a:eGFP);Tg(kdrl:mCherry)* adult tectal meninges; GFP vs. RFP (mCherry) plot showing gates used for cell collection. (c–f) Plots showing relative expression levels of selected genes in FACS sorted FGPs (EGFP-positive, n=~1000 cells per replicate from a total of ~8000 cells sorted) compared to whole fish. (c) Blood endothelial (left) and lymphatic endothelial (right) markers and factors; (d) macrophage, neutrophils and HSPC markers; e) glial and neurotrophic factors; (f) pericyte markers. Relative expression is plotted on a log two scale. Scale bars: 500 μm (a).

The following figure supplement is available for figure 6:

**Figure supplement 1.** Analysis of gene expression in FACS-sorted adult zebrafish FGPs compared to kdrl:mCherry-positive blood endothelial cells.

they lose their migratory behavior and become sessile, although they appear to maintain some protrusive behavior throughout life. We performed additional time-lapse imaging experiments focusing on the region behind the eye from which FGPs appeared to be emerging. Although visualization of the deep cranial regions behind the eye is challenging, time-lapse imaging in *Tg(mrc1a:eGFP);Tg(kdrl:mCherry)* animals shows that FGPs emerge from the optic choroidal vascular plexus (OCVP), a network of vessels that forms adjacent to the pigment epithelium on the deep medial surface of the eye (*Figure 7a*, *Video 3 and 4*). FGPs delaminate from the OCVP and begin to express high levels of Mrc1a:eGFP, eventually migrating away from the OCVP, generally along Kdrl:mCherry-positive vessels (*Figure 7f–j*).

To further investigate whether FGPs emerge from OCVP endothelium, we carried out additional early photoconversion experiments targeting the OCVP using the same *Tg(egfl7:gal4);Tg(UAS:kaede);Tg(mrc1a:eGFP)* endothelial specific triple transgenic line (*Figure 7k–r*) and *Tg(fli:gal4);Tg(UAS:kaede);Tg(mrc1a:eGFP)* endothelial and neural crest specific triple transgenic line (*Figure 7— figure supplement 2*) we used to photoconvert the dorsal aorta. Photoconversion of the OCVP was carried out at 2.5 dpf, before FGP emergence, and the animals were scored for the presence of red fluorescence in FGPs on the optic tectum at 4–5 dpf. Unlike photoconversion of the dorsal aorta, which did not result in any red fluorescent labeling of brain FGPs, photoconversion of the OCVP led to red fluorescent labeling of FGPs in both triple-transgenic lines (*egfl7:gal4* line: 1.25 ± 1.14 S.D.

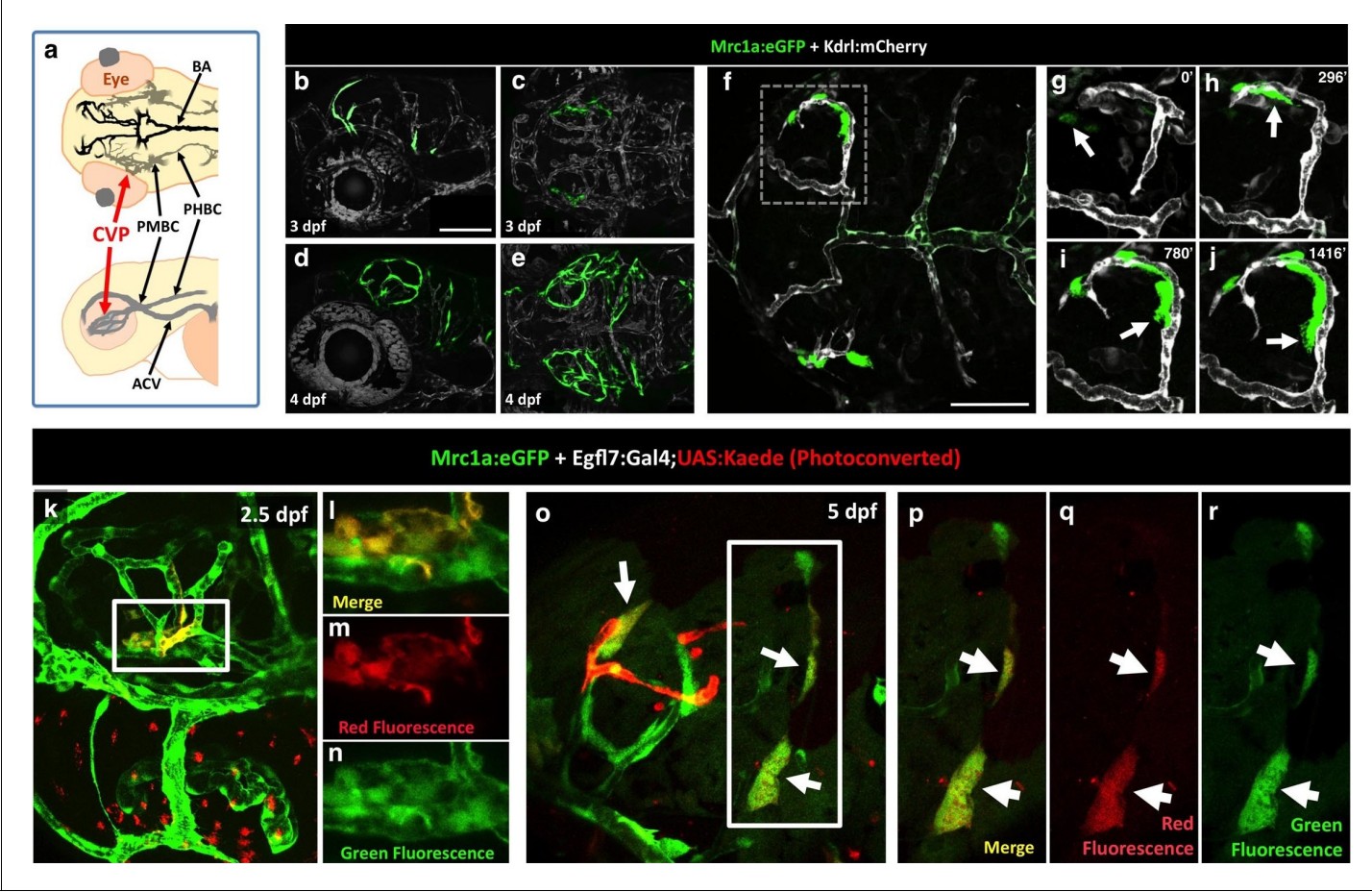

**Figure 7.** Zebrafish FGPs emerge from the endothelium of the choroidal vascular plexus. (**a**), Dorsal (top) and lateral (bottom) view schematic diagrams of approximately 2.5 dpf zebrafish heads with some of their associated vasculature, especially major venous tracts. CVP, choroidal (optic choroidal) vascular plexus; PMBC, primordial midbrain channel; PHBC, primordial hindbrain channel; BA, basilar artery. Adapted from *Figure 5B* in the Vascular Anatomy of Zebrafish Atlas (*Isogai et al., 2001*) – see this reference for additional details. (**b–e**) Confocal images of Mrc1a:eGFP-positive FGPs (green) and Kdrl:mCherry-positive blood vessels (grey) on the surface of the brain in 3 dpf (**b,c**) and 4 dpf (**d,e**) *Tg(mrc1a:eGFP);Tg(kdrl:mCherry)* double-transgenic zebrafish. b and d are lateral views, c and e are dorsal views of the head with rostral to the left. Some residual blood vessel GFP fluorescence was deleted for clarity; see *Figure 7—figure supplement 1* 4a–d for the original unmanipulated green/magenta images. (**f–j**) Confocal time lapse imaging of Mrc1a:eGFP-positive (green) FGPs migrating dorsally then medially along Kdrl:mCherry-positive (magenta) blood vessels on the surface of the brain in a 2.5–3 day old *Tg(mrc1a:eGFP);Tg(kdrl:mCherry)* double-transgenic zebrafish. Panel **f** shows an overview dorsal image from the 780' minute time point (**h**). Panels g-j show magnified views of the boxed region in panel f at 0, 296, 780, and 1416 min time points. (**k–n**) Dorso-lateral view of a 2.5 dpf *Tg(egfl7:gal4);Tg(UAS:Kaede);Tg(mrc1a:eGFP)* triple-transgenic embryo readily after photoconverting the OCVP (k, white box). Red dots in k represent eye autofluorescent pigment. (**l-n**), Higher magnification of the photoconverted OCVP depicted in k (white box) showing expression of red photoconverted Kaede (l,m) and GFP (l,n). (**o**) Photoconversion on the OCVP results in red Kaede Mrc1a:eGFP-positive cells. Lateral view of a 5 dpf *Tg(egfl7:gal4); Tg(UAS:Kaede); Tg(mrc1a:eGFP)* triple-transgenic embryo (**o**) showing Red Kaede, Mrc1a-GFP-positive FGPs (white arrows) on the optic tectum (n = 8/12 embryos showed Red Kaede, with 1.25 ± 1.14 S.D. red FGPs per tectal neuropile).( **p–r**), Higher magnification views of the boxed region in panel o showing Mrc1a:eGFP (green, panels p and r) and photoconverted Kaede (red, panels p and q) double positive FGPs (white arrows). Rostral is to the left in all panels. Scale bars = 100 μm.

The following figure supplements are available for figure 7:

**Figure supplement 1.** Emerging FGPs on the 3 and 4 dpf zebrafish brain.

**Figure supplement 2.** Zebrafish FGPs emerge from the endothelium of the choroidal vascular plexus.

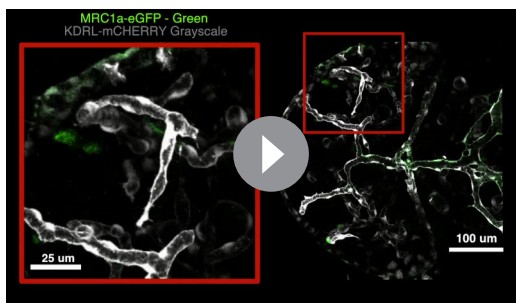

**Video 2.** Confocal time-lapse movie (dorsal view) of developing Fluorescent Granular Perithelial cells migrating to the surface of the brain in a *Tg(mrc1a: eGFP);Tg(kdrl:mCherry)* double transgenic zebrafish embryo from 52 hpf to 80 hpf. *Tg(mrc1a:eGFP)* displayed in green, *Tg(kdrl:mCherry)* displayed in gray scale. Frames acquired every 12 min using a 40 × 1.1 NA water objective with a 4-square tile and 10% overlap on a Zeiss LSM880 confocal microscope. Rostral is to the left. All frames are represented as Z-maximum intensity projections.

FGPs per tectal neuropile, n = 12 embryos; *fli: gal4* line = 1.37 ± 1.98 S.D. FGPs per tectal neuropile, n = 19 embryos), together with some adjacent blood vessels that grew dorsally as well (*Figure 7o–r*, *Figure 7—figure supplement 2b–e*).

Time lapse imaging of FGPs emerging from the OCVP in *Tg(mrc1a:eGFP);Tg(kdrl:mCherry)* animals shows that nascent FGPs are weakly Kdrl: mCherry positive, as might be expected of cells differentiating from primitive endothelium (*Figure 8a–g*, *Video 5*), however this red fluorescence diminishes and is lost as the cells migrate away from the OCVP. We were able to visualize Kdrl-positive emerging FGPs even more clearly by performing time-lapse imaging of the OCVP in *Tg(mrc1a:eGFP);Tg(kdrl:nlsmCherry)* animals with red fluorescent endothelial nuclei (*Figure 8h–k*, *Video 6*), where Kdrl:nlsmCherry-positive FGPs were readily observed migrating out from the OCVP.

The endothelial origin of FGPs was further confirmed using a *Tg(kdrl:Cre)[s898];Tg(-9.8actb2: LOXP-DsRed-LOXP-EGFP)[s928]* (*Kikuchi et al., 2010*) double transgenic 'switch' line (*Figure 9a–c*). In this double transgenic line, non-endothelial tissues are all dsRed-positive and EGFP-negative, but a small percentage of endothelial cells mosaically become EGFP-positive (Cre expression in the Kdrl:Cre line is not sufficient to switch EGFP on in all endothelial cells). As expected, EGFP-positive blood vessels were observed in the double-transgenic line at 5 dpf (*Figure 9d*, *Figure 9—figure supplement 1a–b*), although only a minor subset of endothelial cells were 'switched' to red fluorescence, as seen clearly by comparing with complete vessel labeling in a *Tg(kdrl:mCherry)* embryo (*Figure 9—figure supplement 1c*). Using Qdot705 angiography to mark developing brain blood vessels in 4 dpf animals, we also observed a subset of EGFP-positive FGPs adjacent to these vessels (*Figure 9d,e*). As similar small proportion of

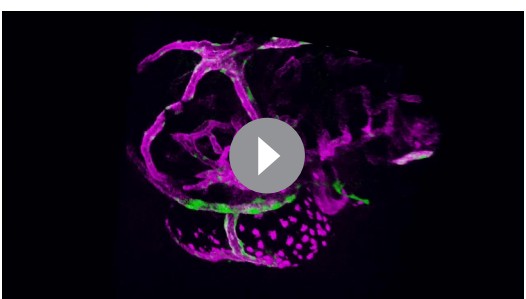

**Video 3.** Confocal time-lapse movie of developing Fluorescent Granular Perithelial cells around the brain of a *Tg(mrc1a:eGFP);Tg(kdrl:mCherry)* double transgenic zebrafish embryo from 52 hpf to 71hpf. *Tg (mrc1a:eGFP)* displayed in green, *Tg(kdrl:mCherry)* displayed in magenta. The movie begins with a dorsal to ventral 3D rotation and ends with a ventral to dorsal 3D rotation. Stacks were acquired every 5.7 min using a 20X, 1.0NA objective on a Leica SP5II confocal microscope. Rostral to the left.

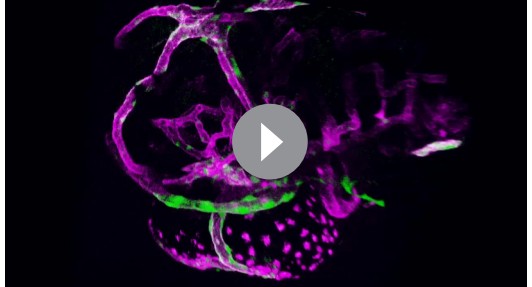

**Video 4.** Confocal time-lapse movie of developing Fluorescent Granular Perithelial cells around the brain of a *Tg(mrc1a:eGFP);Tg(kdrl:mCherry)* double transgenic zebrafish embryo from 52 hpf to 71hpf. *Tg (mrc1a:eGFP)* displayed in green, *Tg(kdrl:mCherry)* displayed in magenta. Stacks were acquired every 5.7 min using a 20X, 1.0NA objective on a Leica SP5II confocal microscope. Rostral to the left. The movie sequence is the same as in *Video 2*, but without the 180-degree dorsal to ventral rotation.

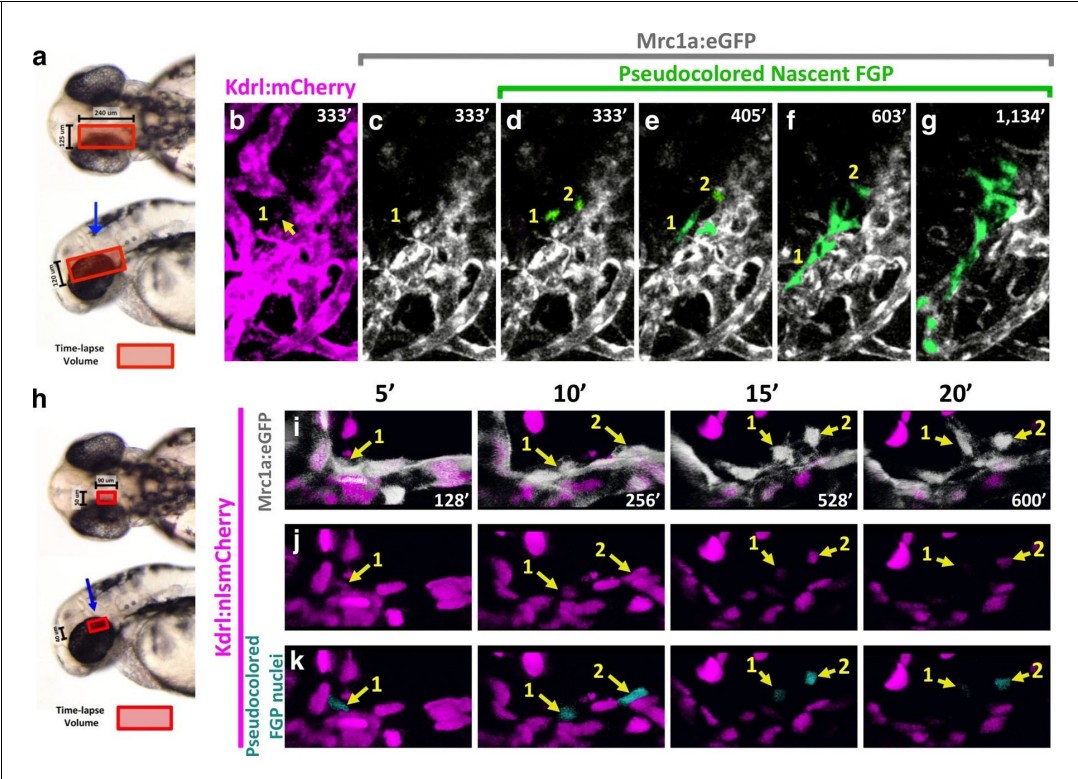

**Figure 8.** Zebrafish FGPs differentiate from the CVP endothelium. (a) Model diagram showing the location of the CVP imaged during timelapse acquisition in a dorsal (top) and lateral (bottom) orientation.( b–g) Stills from time-lapse movies of cells emerging from the CVP at around 2.5–3 dpf in a *Tg(mrc1a:eGFP);Tg(kdrl:mCherry)* double transgenic animal. Mrc1a:eGFP vessels are depicted in gray for ease of visualization, Kdrl:mCherry vessels are magenta and emerging FGPs are green (yellow arrows and numbers 1,2). Mrc1a:eGFP-positive FGPs emerge as single cells and stay in contact with the vessels in the perivascular space. (h), Model diagram showing the location of the CVP imaged during timelapse acquisition in a dorsal (top) and lateral (bottom) orientation.( i–k) Emergence of FGPs from the CVP in Tg(*mrc1a:eGFP*);Tg(*kdrl:nlsmCherry*) double transgenic animal. Newly emerging FGPs (white, yellow arrows, numbers 1–2) express Kdrl:nlsmCherry as they detach from the CVP (yellow arrows in j and teal pseudocolor in k).

EGFP-positive blood vessel endothelial cells and EGFP-positive FGPs were observed on the brains of *Tg(kdrl:Cre)*[s898];*Tg(-9.8actb2:LOXP-DsRed-LOXP-EGFP)*[s928] double transgenic adult animals (*Figure 9f–i*, *Figure 9—figure supplement 1d–f*). The presence of EGFP-positive FGPs in this transgenic switch line confirms the idea that FGPs emerge from Kdrl-positive progenitors early in development. Together, these and our other results suggest that zebrafish brain FGPs have a primitive venous endothelial origin.

## Discussion

In this study, we present the first description of a zebrafish brain perivascular cell population analogous to Fluorescent Granular Perithelial cells (FGPs) of the mammalian CNS. In addition to sharing the same anatomical location in the brain meninges, the zebrafish cells recapitulate other features of mammalian FGPs, notably numerous autofluorescent internal vesicles and the ability to take up foreign or toxic particles from the extracellular space. Although previous studies have characterized FGPs as 'macrophage-like,' our findings suggest that these cells are not of hematopoietic origin but instead are derived from the endothelium of the optic choroidal vascular plexus, a primitive endothelial vascular plexus that resides deep within the brain behind the eyes, from where FGPs detach and migrate to populate blood vessels on the surface of much of the brain.

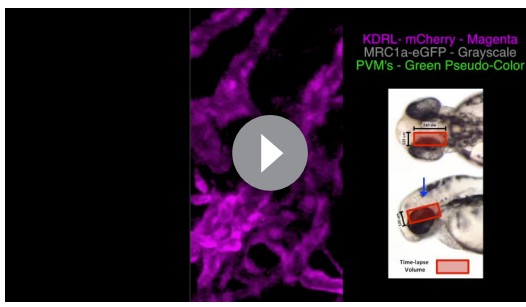

**Video 5.** Confocal time-lapse movie of developing Fluorescent Granular Perithelial cells emerging from the optic choroidal vascular plexus of a *Tg(mrc1a: eGFP);Tg(kdrl:mCherry)* double transgenic zebrafish embryo from 52 hpf to 71 hpf. *Tg(mrc1a:eGFP)* displayed in gray scale, *Tg(kdrl:mCherry)* displayed in magenta. Nascent FGP's are pseudocolored in green. The red square illustrates the confocal volume. The blue arrow indicates the angle of acquisition. Stacks were acquired every 9 min using a 20X, 1.0NA objective on a Leica SP5II confocal microscope. Rostral to the left.

## FGPs, Mato cells, or PVMs?

As noted above, FGPs or 'Mato Cells' were first described by Masao Mato in 1979 (*Mato and Ookawara, 1979*) as a specialized form of PVM closely associated with blood vessels in the lepto-meninges and the cerebral cortex of mammals. They have since been well-described morphologically and ultrastructurally in the mouse, rat, and human brain (*Mato and Ookawara, 1979*; *Mato et al., 1981*, *1986b*, *1982a*, *2002*; *Yokoo et al., 2000*). Commonly known as FGPs due to their strongly auto-fluorescent internal vesicles, these cells are believed to play a scavenger function, protecting the brain from potentially toxic waste products by phagocytosis and pinocytosis (*Mato et al., 1997*, *1996*, *1982b*, *1984*, *2002*; *Mendes-Jorge et al., 2009*). Their macrophage-like morphology has also led to these cells being described as 'perivascular macrophages' (PVMs); indeed the term 'PVM' is currently used much more widely in the literature than either FGP or Mato Cell, although all of the terms have been used somewhat interchangeably in many reports (*Williams et al., 2001*). However, FGP-like cells with fluorescent vesicular inclusions have only been reported on the brain, while PVM-like cells have been reported in a variety of other tissues in addition to the CNS. At this point it is still unclear whether PVMs and FGPs are the same cells, or if FGPs represent a specialized brain sub-population of PVMs. In mouse ear skin, cells described as PVMs secrete chemokines to attract neutrophils during infection or injury (*Abtin et al., 2014*), and in the murine mesentery PVMs influence vascular integrity by limiting permeability (*He et al., 2016*). Although skin and mesenteric PVMs share at least some morphological and molecular characteristics with brain FGPs, including robust expression of the Mannose Receptor 1 protein, their relationship to FGPs needs to be further explored. Brain FGPs have been studied thus far primarily at the ultrastructural level, via electron micrographs of brain samples exposed to heavy metals, lipids and other toxic molecules (*Mato et al., 1984*). Molecular and genetic analysis *in vivo* or *in vitro* is still largely lacking, leaving a great deal of uncertainty as to the range of functional roles of these cells, and whether they share some of the more detailed characteristics and functions of the 'PVMs' reported outside the CNS. Since these cells are found only on the surface of the brain, they may play a role in BBB integrity in the meninges, but not in the deeper brain vasculature from which they appear to be largely absent. Intriguingly, we frequently observed migratory neutrophils physically interacting with FGPs adjacent to 'injured' vascular segments in the zebrafish (*Figure 4h–n*), hinting that zebrafish FGPs may also play a role in

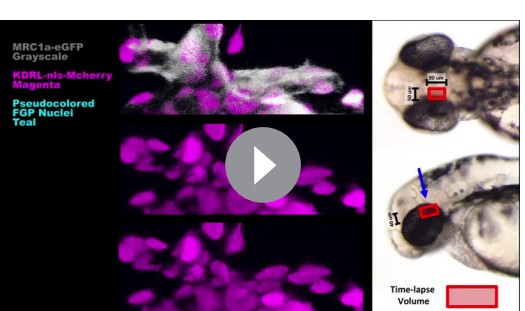

**Video 6.** Confocal time-lapse movie of developing Fluorescent Granular Perithelial cells emerging from the optic choroidal vascular plexus of a *Tg(mrc1a: eGFP);Tg(kdrl:nlsmCherry)* double transgenic zebrafish embryo from 50 hpf to 60 hpf. *Tg(mrc1a:eGFP)* displayed in gray scale, *Tg(kdrl:nlsmCherry)* displayed in magenta, FGP nuclei are pseudo-colored in teal. The red square illustrates the confocal volume. The blue arrow indicates the angle of acquisition. Stacks were acquired every 8 min using a 40X water, 1.1NA objective with an airyscan detector to improve the signal to noise ratio on a Zeiss LSM880 confocal microscope. Rostral to the left.

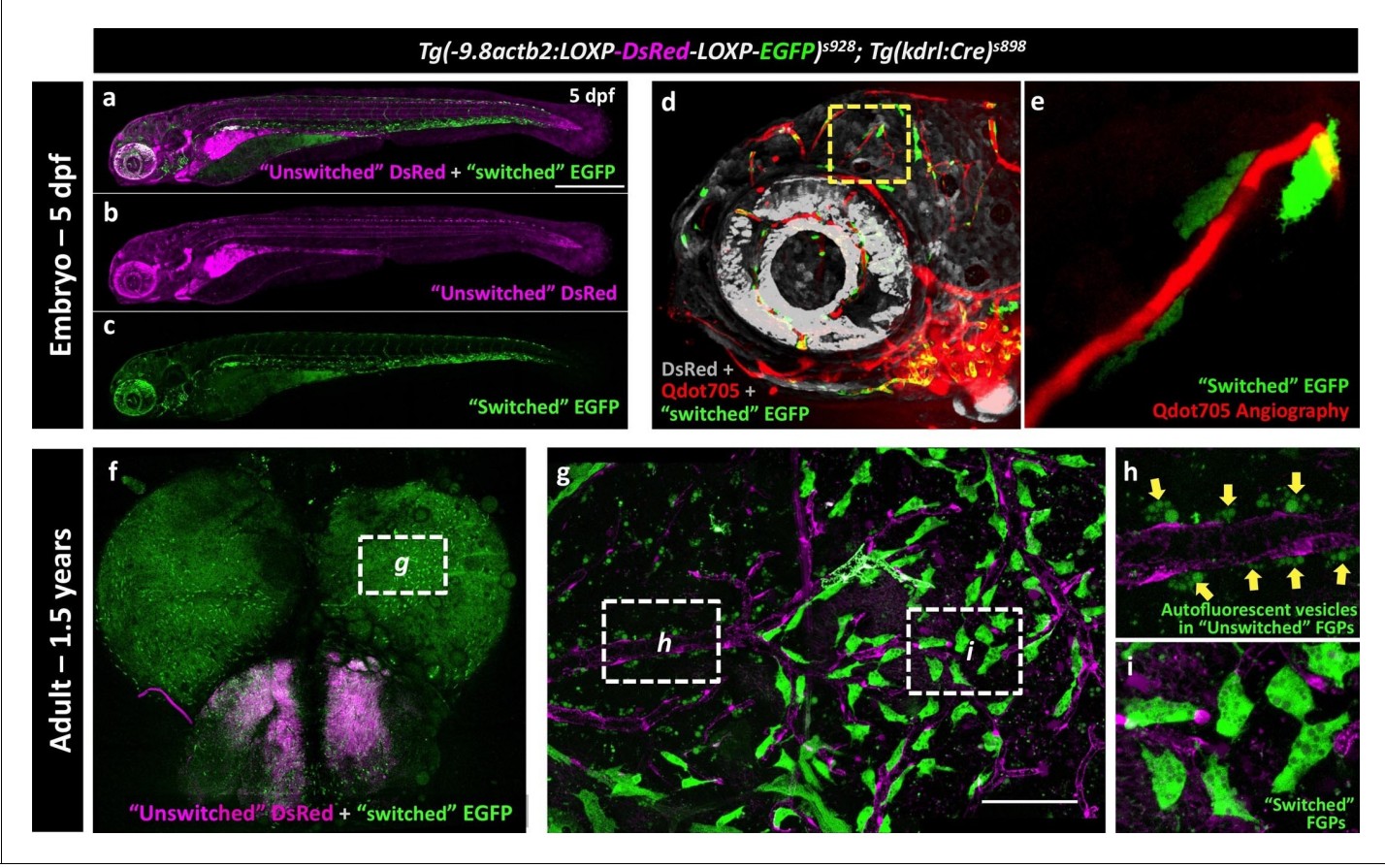

**Figure 9.** Zebrafish FGPs are derived from *kdrl*-expressing endothelium. (a-c) Lateral view confocal micrographs of a 5 dpf *Tg(−9.8actb2:LOXP-DsRED-LOXP-EGFP); Tg(kdrl:Cre)* double transgenic 'switch' embryo, with 'unswitched' DsRed-positive cells (magenta, panels a and b) and 'switched' EGFP-positive cells (green, panels a and c) where Kdrl:Cre has successfully excised the LoxP cassette. (d), Lateral view confocal micrograph of a 4 dpf *Tg(-9.8actb2:LOXP-DsRED-LOXP-EGFP);Tg(kdrl:Cre)* double transgenic embryo injected intravascularly with Qdot705 (red) to highlight all patent vessels. 'Unswitched' DsRed is in grey, and 'switched' EGFP is in green. (e) Higher magnification image of the yellow boxed area in panel d, showing an optic tectum vessel labeled with Qdot705 (red) surrounded by switched EGFP-positive FGPs (green). (f), Dorsal view confocal micrograph of the brain optic lobes of an adult *Tg(−9.8actb2:LOXP-DsRED-LOXP-EGFP); Tg(kdrl:Cre)* double transgenic 'switch' animal with mosaic expression of EGFP in a subset of FGPs. The white box notes the area shown in panel g. (g), Higher magnification image of the boxed region in panel f, with boxes noting the areas shown at still higher magnification in panels h and i. (h) Higher magnification confocal image showing autofluorescent green vesicles (yellow arrows) in unswitched FGPs. (i) Higher magnification confocal image showing EGFP-positive switched FGPs. Rostral is to the left in panels a-e, and up in panel f-i. Scale bars: 500 μm (a–c), 100 μm (g).

The following figure supplement is available for figure 9:

**Figure supplement 1.** Mosaic expression of EGFP in vessel endothelial cells in the Kdrl:Cre 'switch' double transgenic line.

neutrophil recruitment in response to injury. However, further molecular and functional study in genetically and experimentally accessible model organisms such as the mouse and the zebrafish will be needed to clarify the function of these cells, and determine whether FGPs and PVMs are the same type of cell or different perivascular cell types with distinct functions in different tissues.

## Zebrafish PVMs are FGPs

Our analysis of Mrc1a:eGFP-positive cells in the zebrafish brain strongly suggests that these are the same cells as the FGPs of the mammalian brain. Zebrafish and mammalian FGPs share an anatomical perivascular location in the brain meninges, they both express Mannose Receptor 1, a characteristic marker of these cells in mammals, and they both contain internal autofluorescent vesicles, a unique and characteristic feature of FGPs/Mato cells (*Mato et al., 1981*; *Ookawara et al., 1996*). We

carried out a variety of different experiments to rule out the possibility that brain Mrc1a:eGFP-positive perivascular cells represent, or overlap with, other previously characterized brain perivascular cell types. Mrc1a-positive cells are not microglia based on our Neutral Red staining (*Figure 4d,e*), a result supported by findings of previous studies in rats where both cell populations where clearly separated by their time of appearance, anatomical location, morphology and capacity to take up Horseradish Peroxidase (*Mato et al., 1985*). Furthermore, in human brain samples from individuals suffering from neurodegeneration, lymphomas, sarcomas and brain hemorrhages, the marker GP-3 was found to specifically label FGPs and not macrophages or microglia under these immune compromised conditions (*Yokoo et al., 2000*). Similarly, pericytes and FGPs were also described as separate cell populations in rat cerebrums, again based on location and distribution (*Ookawara et al., 1996*). These cells are also morphologically distinct in electron micrographs, with FGPs appearing swollen and loaded full of granules, whereas pericytes were flat (*Ookawara et al., 1996*). Confocal imaging of *Tg(lyve1:dsRed);Tg(pdgfrb:Citrine)* double-transgenic adult zebrafish brains confirmed that Mrc1a:eGFP-positive cells do not overlap with pericytes or share expression of Pdgfrb, although the two cell types are located in very close proximity to one another on the brain vasculature (*Figure 4b,c*). Finally, zebrafish brain pericytes do not show inclusions and are thinner and smaller than FGPs. In a similar manner, we were able to show that Mrc1a:eGFP-positive cells do not correspond to Mpx:eGFP-positive (*Figure 4f,g*) or Lyz:dsRed-positive neutrophils (*Figure 4h–n*), Mpeg1-positive macrophages (*Figure 4o–t*), or Gfap:GFP-positive glia (*Figure 4—figure supplement 1*). Together, the characteristic features shared by Mrc1a:eGFP-positive cells and the lack of correspondence to other characterized perivascular cell types leads to the conclusion that these cells represent the zebrafish equivalent of mammalian FGPs or Mato cells. The availability of a new zebrafish model for studying FGPs will provide an extremely valuable new resource for experimental characterization of these cells and their functional role in the brain.

## An early endothelial origin for zebrafish FGPs

Although previous reports have suggested a hematopoietic origin for FGPs, a number of our findings do not support this conclusion. In all vertebrates examined, definitive hematopoietic cells emerge from endothelial cells in the floor of the dorsal aorta that transdifferentiate into hematopoietic stem and progenitor cells (HSPCs) and delaminate from the vessel wall before migrating to take up residence in hematopoietic organs. In the zebrafish, primitive hematopoiesis is triggered as early as 14 hpf and definitive HSPC-dependent specification and delamination occurs in the dorsal aorta at approximately 2–3 dpf (*Boisset et al., 2010*; *Gérard et al., 2010*; *Kissa and Herbomel, 2010*; *Murayama et al., 2006*). Blocking primitive hematopoiesis by knocking down *pu.1* did not affect the formation of FGPs (*Figure 5—figure supplement 1*). By 'tagging' definitive HSPCs before they emerged from the dorsal aorta, using transgenic lines expressing green-to-red photoconvertible Kaede protein in endothelial cells, we were able to determine which cell types and tissues these cells contributed to at later stages. As expected, cells with photoconverted Kaede were readily detected in the 5 dpf thymus, a hematopoietic organ 'seeded' by HSPC-derived cells, as well as in trunk endothelium, but no photoconverted Kaede could be detected in Mrc1a:eGFP-positive FGPs (*Figure 5b–h*). Together, these results suggest that FGPs are not derived from either hematopoietic wave. Zebrafish FGPs are also not marked by Runx1:GFP and c-Myb:eGFP transgenes that mark other hematopoietic-derived cells (*Figure 5i–n*), and as noted above they fail to express either Mpx:eGFP or Lyz:dsRed neutrophil and Mpeg1-mCherry macrophage-specific transgenes (*Figure 4f–t*). Importantly, we also found no change in either the number or localization of FGPs in *runx1*-deficient zebrafish, despite strong reduction in the number of Mpx-positive neutrophils in the same animals (*Figure 5—figure supplement 3*). Taken together, these results strongly suggest a non-hematopoietic origin for these cells.

In addition to FGPs, the Mrc1a:eGFP transgene is strongly expressed in lymphatic endothelial cells (*Jung et al., 2017*). Surprisingly, we found that other transgenes used to visualize lymphatics in the zebrafish are also strongly expressed in zebrafish FGPs, including Lyve1:dsRed and Prox1a:RFP (*Figure 2h–m*). The latter is of particular interest because Prox1 is a transcription factor that has sometimes been called the 'master regulator' of lymphatic development in mammalian and avian models (although it is expressed in other non-vascular tissues), and its expression and that of Lyve1 are characteristic of lymphatic as well as primitive venous endothelium (*Burke and Oliver, 2002*; *Dyer et al., 2003*; *Hong and Detmar, 2003*; *Hong et al., 2002*; *Johnson et al., 2008*; *Lee et al.,*

*2009*; *Okuda et al., 2012*; *Risebro et al., 2009*; *Rodriguez-Niedenführ et al., 2001*; *Shin et al., 2006*; *Sosa-Pineda et al., 2000*; *Wigle and Oliver, 1999*). Using a comprehensive RNA-seq analysis of the genes expressed in FACS-sorted adult brain FGPs, we showed that these cells express elevated levels of a variety of lymphatic or lympho-venous markers including *mrc1a, lyve1b,* and *prox1b,* and the lymphangiogenic ligand *vegfd* (*Alders et al., 2009*; *Astin et al., 2014*; *Hogan et al., 2009*; *Lee et al., 2009*; *Okuda et al., 2012*; *van Impel et al., 2014*), when compared to either whole fish (*Figure 6*) or to co-sorted blood endothelial cells (*Figure 6—figure supplement 1*). Interestingly, FGPs show strongly reduced expression of several key blood endothelial genes including *kdrl, pecam1,* and *vegfaa* (*Baluk et al., 2007*; *Covassin et al., 2009*; *Sauteur et al., 2014*; *Wang et al., 2010*) – the reduced *kdrl* (*vegfr2* receptor) expression confirms the lack of *kdrl:mCherry* transgene expression observed in adult FGPs and highlights the distinction between FGPs and blood endothelial cells. Examination of a variety of hematopoietic, macrophage, astrocyte/glial, neurotrophic factors and pericyte genes (*Gore et al., 2016*; *Hall et al., 2007*; *Herbomel et al., 2001*; *Kissa and Herbomel, 2010*; *Lam et al., 2010*; *Le Guyader et al., 2008*; *Sood et al., 2010*) also confirms that the gene expression profile of FGPs does not resemble any of these cells types (*Figure 6d–f*, *Figure 6—figure supplement 1b–d*). The RNA-seq data also provides a unique gene expression signature for this unique cell type that may also be useful for future studies.

Together, these findings led us to hypothesize that FGPs might be derived from early lymphatic or lympho-venous endothelium. Indeed, we showed using time-lapse imaging of double–transgenic animals that FGPs emerge from the optic choroidal vascular plexus, a network of venous vessels located inside the brain behind the eyes, a result confirmed by labeling of FGPs after photoconversion of this vascular plexus in *Tg(egfl7:gal4ff); Tg(UAS:Kaede); Tg(mrc1a:eGFP)* or *Tg(fli:gal4);Tg (UAS:Kaede);Tg(mrc1a:eGFP)* triple transgenic animals (*Figure 7k–r* and *Figure 7—figure supplement 2a–e*). Using time-lapse imaging of FGPs budding from the optic choroidal vascular plexus in *Tg(mrc1a:eGFP);Tg(kdrl:nlsmcherry)* double transgenic animals we were also able to show that nascent FGPs are initially Kdrl-positive as they emerge from this plexus and began to migrate away (*Video 6*). Finally, we used a double transgenic *Tg(kdrl:Cre)^{s898};Tg(−9.8actb2:LOXP-DsRed-LOXP-EGFP)^{s928}* 'switch' line to show that FGPs are lineage-marked by kdrl-expression like normal endothelial cells. The *kdrl* gene encodes a highly vascular endothelial-specific transcription factor (*Covassin et al., 2006*), and the Kdrl:nlsmCherry and Kdrl:Cre transgenes are expressed exclusively in endothelial cells, so the transient appearance of Kdrl:mCherry in FGP nuclei as they emerge from the optic choroidal vascular plexus and permanent marking of FGPs in the Kdrl:Cre background provides additional strong evidence that these cells are transdifferentiating from blood vascular endothelium. The idea that endothelial cells are capable of differentiating into alternative cell types is of course not without precedent. As noted above, the HSPCs that give rise to the definitive hematopoietic system differentiate from endothelial cells in the floor of the dorsal aorta (*Kissa and Herbomel, 2010*; *Murayama et al., 2006*). Although FGPs have been described as 'macrophage like,' the lack of molecular similarity to macrophages or other hematopoietic cell types suggests that the parallels between HSPC specification from the dorsal aorta and FGP emergence from the optic choroidal vascular plexus are limited. It is worth noting that while HSPCs rapidly lose endothelial identity after emerging from the dorsal aorta, FGPs maintain a gene expression signature of lympho-venous identity indefinitely.

## Conclusions and perspectives

Although our study provides a solid foundation, further work will be needed to elucidate the functional role of FGPs in vascular growth, vascular function, blood brain barrier formation and function, and brain homeostasis and responses to pathological insults. Further studies will also be needed to fully characterize the emergence of these cells from the primitive endothelium of the optic choroidal vascular plexus, and the molecular mechanisms that underlie the specification of this novel cell type. Although as noted above there may be some parallels between the emergence of HSPCs from the dorsal aorta and the emergence of FGPs from the optic choroidal vascular plexus, the lack of *runx1* or *c-myb* expression in nascent FGPs and the lack of functional consequences for production of FGPs resulting from *runx1* knockdown suggests that distinct molecular mechanisms are involved. Further exploration of the molecular basis for FGP specification should be enlightening. With its genetic and experimental accessibility, and optically clear embryos and larvae that facilitate high-

resolution light microscopic imaging, the zebrafish provides a superb model for further functional characterization of this novel and unusual perivascular cell type.

## Materials and methods

### Fish husbandry and fish strains

Fish embryos were raised in E3 media and kept at 28.5°C until the desired developmental stages. The following lines were used for this study: *Tg(mrc1a:eGFP)$^{y251}$* (*Jung et al., 2017*), *Tg(fliep:gal4f-f)$^{ubs4}$* (*Totong et al., 2011*), *Tg(UAS:kaede)* (*Hatta et al., 2006*), *Tg(mpx:eGFP)$^{i114}$* (*Renshaw et al., 2006*), *Tg(−5.2lyve1b:DsRed)$^{nz101}$* (*Okuda et al., 2012*); *Tg(prox1aBAC:KalTA4-4xUAS-E1b:uncTagRFP)$^{nim5}$* (*van Impel et al., 2014*), *Tg(lyz:dsRed)* (*Hall et al., 2007*), *Tg(flk:mCherry)* (*Wang et al., 2010*), *Tg(c-myb:eGFP)* (*North et al., 2007*), *Tg(kdrl:Cre)$^{s898}$* (*Bertrand et al., 2010*), *Tg(-9.8actb2:LOXP-DsRed-LOXP-EGFP)$^{s928}$* (*Kikuchi et al., 2010*), *Tg(mpeg1:Gal4-VP16)* (*Ellett et al., 2011*) and *TgBAC(pdgfrb:citrine)$^{s1010}$* (*Vanhollebeke et al., 2015*). The *Tg(runx1:eGFP)$^{y509}$* transgenic line was generated by cloning the *runx1* enhancer/promoter sequence (*Tamplin et al., 2015*) upstream of EGFP into a Tol2 construct using Gateway technology (*Kwan et al., 2007*). The *gSAlzGFFD478A* (*Tg(egfl7:gal4ff)$^{gSAlzGFFD478A}$*) line was made by the Tol2-transposon mediated gene trap method as previously described (*Kawakami et al., 2010*).

### Microangiography, lymphoangiography and intracranial injections

Microangiography and lymphangiography in developing animals was performed using Qdot705 as previously described (*Yaniv et al., 2006*). For lymphangiography (fluorescent dye uptake into and drainage through the lymphatics), undiluted Qdot705 were injected into the dorsal side of the brain or the trunk musculature, while for angiography Qdot705 were injected directly into the caudal posterior cardinal vein using a filamented glass needle. Intracranial injections of India ink into adult animals were performed on 10 one-year old zebrafish anesthetized with MS-222 (4 g/mL) and positioned ventrally on a dissecting mat. 10 nL of pure India ink was injected using capillary glass needles at the most posterior border of the skull bony plates. India ink injected adult animals were allowed to recover in fresh fish water for 24 hr and then the fish were euthanized on ice and their brains dissected and imaged.

### Tissue sectioning

Adult zebrafish were euthanized on ice and their brains were dissected out manually and washed in 1X PBS. For cryosectioning, brains (n = 3) were embedded in O.C.T Compound media (Tissue-Tek) and sectioned into 10 μm thick sections (Histoserv, Inc.) and stained with an anti-GFP primary antibody (1:2000). Adult mice (n = 3) were anesthetized with Isoflurane and perfused intracardially with 50 mL of 1X PBS, followed by 50 mL of 4% PFA. The brains were mechanically dissected and place in 1X PBS for a wash, followed by a gradient of 5%, 10% and 20% Sucrose/PBS washed with 0.01% Sodium Azide. Brains were then cryosectioned into 50 μm slices and stained with the Anti-Mannose Receptor CD206 antibody (ab64693, Abcam - Rabbit polyclonal) and DAPI.

### Immunohistochemistry

Cryosections of mouse or zebrafish brains on slides were pre-warmed at RT in TBS solution and then washed three times in fresh TBS to remove all O.C.T., leaving behind the sections on the slides. Staining was performed as follows: sections were incubated 30 min in 1X Tris Glycine, then 1 hr in 0.1% Triton X-100 and finally 1 hr in blocker (1% Roche Blocking Buffer, 5% Sheep Serum in TBS-T). Samples were incubated overnight at 4°C in blocker with primary antibody (GFP 1:2000 and CD206 at 1:1000). Samples then were washed in TBS-T, four or more times as needed and incubated in blocker with secondary antibody for up to 2 hr at room temperature and washed in TBS-T at least four times.

### Whole mount *in situ* hybridization and imaging

In situ hybridization was performed according to standard protocols, with minor modifications (*Kopinke et al., 2006*). For *mpx* antisense probe (*Bresciani et al., 2014*) staining, embryos were fixed at 20 hpf in 4% PFA overnight at 4°C, dehydrated in ascending MeOH series and stored in

100% MeOH at −20℃ overnight. Embryos were then rehydrated and washed in PBST (1X PBS 0.1% Tween) three times for 5 min and transferred into Hyb buffer for 4 hr at 68℃, followed by antisense probe incubation for 14 hr at 68℃. Samples were then washed in 1:1 deionized formamide-2X SSC buffer 2 times at 65℃, transferred into Hyb wash buffer (50% deionized formamide, 5X SSC, 0.25% CHAPS) and washed 3 times for 30 min, followed by 2 washed of 15 min on Hyb wash/2X SSC (1:1 and 1:3). Embryos then were transferred into 2X SSC-Ch (2X SSC, 0.25% CHAPS) for 15 min, followed by two 30 min washes in 0.2X SSC-Ch (0.2X SSC, 0.25% CHAPS) and a 30 min wash in PBST-Ch (1X PBS 0.1% Tween, 0.25% CHAPS). For antibody incubation, embryos were washed twice for 15 min in 1X MAT (MAB buffer, 0.1% Tween) and blocked for 90 min in MABT (1X MAT, 1 mg/mL BSA), followed by a 14 hr incubation in MABT, 10% NGS and 1:4000 of anti-DIG-AP antibody (Sigma-Aldrich) at 4℃. Finally, embryos were washed 8 times for 20 min in 1X MAB buffer and transferred into staining buffer (100 mM NaCl, 50 mM MgCl, 100 mM Tris-HCL pH9.5, 10% Tween) for 10 min three times. Staining was done at RT with 3.4 uL/mL NBT; 3.5 ul/mL BCIP in staining buffer and stopped with 4% PFA.

## Neutral red staining

4.5 dpf zebrafish *Tg(mrc1a:eGFP)* larvae were incubated in a 2.5 µg/mL Neutral Red vital dye solution in E3 media for 12 hr at 28.5℃ and then imaged live on a Leica Planapo 1.0X stereo scope (n = 15). Control embryos were incubated under the same conditions in E3 media alone (n = 15).

## Imaging

Zebrafish larvae up to 6 dpf were anesthetized with MS-222 and mounted in 0.8% low melting point agarose. Juveniles and adult zebrafish were euthanized on ice and their brains were dissected manually on a dissecting mat, mounted in 2% methylcellulose in a Mat-Tek imaging dish or a glass slide with a cover slip. Brain cryosections were covered in mounting media, cover slipped and sealed with nail polish. Samples were imaged in either a Leica TCS SP5-II upright confocal microscope using a 20X immersion objective, an inverted Zeiss LSM880-Airyscan using a 10X air or 40X long working distance water objective, an upright Olympus confocal microscope, or a Leica stereoscope with Planapo 1.0X objective and an Olympus DP71 camera. Image processing was performed in ImageJ/Fiji, Volocity, or Photoshop. Time-lapse image stacks were cropped in time and volume, and levels were adjusted using Volocity (Perkin-Elmer) and Photoshop CS6 (Adobe). Movies were assembled and labeled using Premier Pro CS6 (Adobe) or Final Cut Pro (Apple). Cell nuclei were pseudo-colored using Photoshop CS6 (Adobe), or After Effects CS6 (Adobe). For in situ staining acquisition, stained embryos were mounted in 0.8% LMP on a Lab-TekII chamber (Borosilicate #1.5 German coverglass – 155379) and imaged on a Nikon Eclipse T*i*2 inverted microscope with to a Nikon DS-R*i*2 camera. The acquired Z-stacks were projected using the Nikon EDF feature.

## Kaede photoconversion

GFP-positive *Tg(fli:gal4); Tg(UAS:Kaede); Tg(mrc1a:eGFP)* or *Tg(egfl7:gal4); Tg(UAS:Kaede); Tg(mrc1a:eGFP)* triple transgenic animals were mounted laterally or dorso-laterally in 0.8% low melting point agarose in a Mat-Tek dish. The entire dorsal aorta or regions of the optic CVP were exposed to a UV laser on a Zeiss LSM880 confocal microscope using a 40X water objective. Dorsal aorta photoconversions were performed at 24–28 hpf and CVP photoconversions were performed at 2.5 dpf. After photoconversion, embryos were manually removed from the agarose and placed in fresh E3 media containing 1X PTU and raised to 4–5 dpf at 28.5℃, when they were scored for Red Kaede and GFP-positive FGPs.

## High fat diet and feeding

For an acute response to high cholesterol diet (HCD), adult zebrafish were starved for 48 hr and then split into a control and an HCD group (n = 5 per group). Control group was fed 0.2 g of regular chow and HCD adults were fed 0.2 g of cholesterol chow in three rounds, every 2 hr. Brains of control and HCD diet fish were dissected six hours after the first feeding and FGP cells were imaged on a confocal microscope. Cell length and number of vesicles were quantified from confocal images using Volocity software.

## Morpholino injections

*Tg(lyve1:dsRed);Tg(mpx:eGFP)* double transgenic zebrafish embryos were injected with 4 ng of a splice blocking *runx1*-fluorescein tagged morpholino at the one cell stage (*Gore et al., 2016*). Injected embryos were allowed to develop at 28.5°C until 5 dpf when they were scored for Mpx: GFP- and Lyve1:dsRed-positive cells. Similarly, *Tg(mrc1a:eGFP)* embryos were injected with 4 ng of *pu.1/spi1b* morpholino (*Rhodes et al., 2005*) at the one cell stage. Embryos were raised at 28.5 C until 20 hpf when half of them were fixed and used for whole mount in situ hybridization; the rest were raised until 5 dpf when they were scored for the presence of FGPs based on GFP expression.

## Tissue isolation, FACS sorting and RNA isolation for RNA sequencing

Two biological samples each consisting of ten adult *Tg(mrc1a:eGFP);Tg(kdrl:mCherry)* double transgenic zebrafish brains were euthanized on ice and their brains were immediately dissected. The exterior membrane of the brain containing Mrc1a:eGFP(+) and Kdrl:mCherry(+) cells was carefully dissected and placed on 1X PBS (pH 7.4, without $Ca^{2+}$ and $Mg^{2+}$). Cells were washed with 1X PBS three times and dissociated in 3 mL of Trypsin-EDTA by gentle pipetting. Dissociated cells were then passed through a 70 µm filter and centrifuged at 4000 rpm for 5 min at room temperature. The pellet was washed with Fluorobite DMEM media containing 1% Fetal Bovine Serum (FBS) and spin down for 3 min at 4000 rpm. The washed pellet was resuspended in 3 mL of Fluorobite DMEM media with 1% FBS. Fluorescent cell sorting was performed on a BD FACS ARIA (Beston Dickinson, Franklin Lakes, NJ). Isolated GFP(+), mCherry(+) and dark cells were pelleted at 2500 rpm for 15 min. For control tissue, the whole adult fish, including the GFP negative brain tissue was exposed to liquid nitrogen and ground in a mortar. RNA from pulverized fish was extracted by Trizol-Chlorophorm-Isopropanol isolation and cleaned on Matrigel columns. For all samples, SMART-Seq v4 Ultra Low Input RNA kit (Takara) was used to prepare cDNA from 1000 GFP(+) and 1000 mCherry(+) cells and 1 ng of whole fish RNA. 10 ng of resulting cDNA library material was tagmented, amplified and indexed with Illumina Nextera DNA library preparation kit. Libraries were characterized on an Agilent Bioanalyzer, and quantified via Qubit dsDNA assay, then sequenced on a NextSeq 500 Illumina instrument. Two technical replicates were sequenced for each of the two biological samples, for a total of four replicates for each sample. An approximate average of 24 million total reads were obtained for whole fish samples, 55 million reads for GFP-positive cells and 65 million total reads for RFP-positive cells, per biological replicate.

## RNA sequencing analysis

Data was demultiplexed using bcl2fastq (Illumina), FASTQ sequencing files were trimmed using Trimmomatic for poly-A/G/C/T sequences and Illumina adapter sequence (*Bolger et al., 2014*). Resulting trimmed read files were aligned to zebrafish (zv10) using RNA-STAR (*Dobin et al., 2013*). Following alignment, transcript reads were quantitated by subread featureCounts and tested for differential expression by DESeq2 (*Love et al., 2014*).

## Acknowledgements

We are grateful to Dr. M Cashel for sharing reagents, Dr. A Chitnis for sharing the *Tg(mpx:eGFP)* line, Dr. P Crosier for the *Tg(lyve1:dsRed)* line, Dr. D Stainier for the *Tg(pdgfrb:Citrine)*, *Tg(-9.8actb2:LOXP-DsRed-LOXP-EGFP)* and the *Tg(kdrl:CRE)* lines, Drs. E Ober and T Piotrowski for the *Tg(prox1:RFP)* line, Jason Sinclair and Dr. S Burgess for *Tg(mpeg1:gal4)* line and Drs. O Tamplin and L Zon for sharing the *runx1* enhancer/promoter construct. We also thank G Marquart and Dr. H Burgess (NICHD) for assistance with the Zebrafish Brain Browser. We thank the NHGRI FACS Core for their excellent cell sorting support and Jamie Diemer and Paul Liu (NHGRI) for facilitating our cell sorting efforts. This work was supported by the intramural program of the Eunice Kennedy Shriver National Institute of Child Health and Human Development, NIH (HD008808, to BMW and Z1A-HD000412-30, to RJM) and the NBRP from AMED, and JSPS KAKENHI (JP15H02370 and JP16H01651) to KK.

## Additional information

### Funding

| Funder | Grant reference number | Author |
|---|---|---|
| Eunice Kennedy Shriver National Institute of Child Health and Human Development | Z1A HD008808 | Marina Venero Galanternik<br>Daniel Castranova<br>Aniket V Gore<br>Hyun Min Jung<br>Amber N Stratman<br>Mayumi F Miller<br>Brant M Weinstein |
| Eunice Kennedy Shriver National Institute of Child Health and Human Development | Z1A HD000412-30 | Nathan H Blewett<br>James Iben<br>Richard J Maraia |
| National Human Genome Research Institute | | Martha R Kirby |
| Japan Agency for Medical Research and Development | | Koichi Kawakami |
| Japan Society for the Promotion of Science | JP15H02370 | Koichi Kawakami |
| Japan Agency for Medical Research and Development | JP16H01651 | Koichi Kawakami |

The funders had no role in study design, data collection and interpretation, or the decision to submit the work for publication.

### Author contributions

MVG, Conceptualization, Data curation, Formal analysis, Investigation, Visualization, Methodology, Writing—original draft, Project administration, Writing—review and editing; DC, Conceptualization, Data curation, Visualization; AVG, NHB, MRK, MFM, Data curation, Methodology; HMJ, Data curation, Methodology, Writing—review and editing; ANS, Data curation, Writing—review and editing; JI, Formal analysis, Methodology; KK, Resources, Methodology, Writing—review and editing; RJM, Resources, Funding acquisition, Methodology, Writing—review and editing; BMW, Conceptualization, Supervision, Funding acquisition, Project administration, Writing—review and editing

### Author ORCIDs

Marina Venero Galanternik, http://orcid.org/0000-0002-0262-1754

Koichi Kawakami, http://orcid.org/0000-0001-9993-1435

Brant M Weinstein, http://orcid.org/0000-0003-4136-4962

### Ethics

Animal experimentation: This study was performed in strict accordance with the recommendations in the Guide for the Care and Use of Laboratory Animals of the National Institutes of Health. All of the animals were handled according to approved institutional animal care and use committee (IACUC) protocols (ASP # 12-039 and 15-017).

## Additional files

### Major datasets

The following dataset was generated:

| Author(s) | Year | Dataset title | Dataset URL | Database, license, and accessibility information |
|---|---|---|---|---|
| Marina Venero Galanternik | 2017 | RNAseq | https://www.ncbi.nlm.nih.gov/geo/query/acc.cgi?acc=GSE97421 | Publicly available at NCBI (accession no: GSE97421) |

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
