## [Decision Letter]

Thank you for submitting your article "A Novel Perivascular Cell Population in the Zebrafish Brain" for consideration by *eLife*. Your article has been reviewed by two peer reviewers, one of whom is a member of our Board of Reviewing Editors, and the evaluation has been overseen by Marianne Bronner as the Senior Editor. The following individual involved in review of your submission has agreed to reveal his identity: Philippe Herbomel (Reviewer #3).

The reviewers have discussed the reviews with one another and the Reviewing Editor has drafted this decision to help you prepare a revised submission.

Summary:

This manuscript describes a new perivascular cell population at the surface of the zebrafish brain, which appears to be the homologue of cells previously described in mice as Fluorescent Granular Perithelial Cells (FGPs) or 'perivascular macrophages' or "Mato cells". The authors identified these cells using a new transgenic line where GFP is expressed under the control of the mannose receptor 1 promoter. The data are of very high quality and the writing polished.

The authors further show that these cells express various markers of lymphatic endothelium; that they do not appear to originate from aorta-derived definitive hematopoiesis; they provide evidence from in vivo imaging and uncaging studies that these FGPs originate from a particular vascular endothelium, the optic choroidal plexus.

Although mostly descriptive, this paper is bound to generate renewed interest in the diversity of cell types associated with brain vasculature including the role of this poorly understood cell type.

Essential revisions:

1) Regarding the origin of the studied cells, given their unusual nature and the mouse data documenting a myeloid origin for CD206+ 'PVMs', the authors should confirm that these cells do not originate from the primitive macrophage lineage or from neural crests.

2) While macrophage origin is a major candidate, to exclude it the authors use as a "macrophage marker" the transgenic line mpx:gfp.… which, in zebrafish, only marks neutrophils and not macrophages. The only macrophage-specific transgenic reporter zebrafish lines available are based on the mpeg1 promoter (following G. Lieschke's lab) or Mfap4 promoter (Tobin lab).

3) The authors investigate definitive hematopoiesis as the source of these cells. But why ignore the primitive wave, born in the yolk sac, which produces macrophages of long-term renewal capacity (at least in mice), and could just as well be a source? Testing its involvement is straightforward, as the standard anti-Pu1 morpholino suppresses formation of this lineage.

4) Another obvious candidate source for the studied FGPs are neural crest cells. It is surprising that the authors do not even discuss this possible origin, especially since their lab addressed a similar issue in a very recently published paper (Stratman et al. Development 2017). Moreover, to prove the endothelial origin of the FGPs, they use photoconversion at 24 hpf of Kaede expressed in the Tg(Fli1:gal4; UAS:Kaede) line. Unfortunately, the fli1a gene is well known to be expressed in ectomesenchymal derivatives of NCCs (there is also primitive macrophage expression, see e.g. the yolk sac in Figure 5 of this manuscript), so that when the authors photoconvert an area medial to the eyes at 24 hpf, they necessarily photoconvert not only vascular but also neural crest cells (and possibly also primitive macrophages), which are numerous in this area at this stage.

5) Is there an endothelial Cre (or Cre-ER) line available that could be used to confirm the endothelial origin of these cells?

---

## [Author Response]

Essential revisions:

1) Regarding the origin of the studied cells, given their unusual nature and the mouse data documenting a myeloid origin for CD206+ 'PVMs', the authors should confirm that these cells do not originate from the primitive macrophage lineage or from neural crests.

In response to these comments, we carried out additional experiments to verify that zebrafish FGPs are not derived from the primitive macrophage lineage or neural crest:

a) As suggested (see below) we blocked primitive hematopoiesis using a previously published ATG pu.1 morpholino (Rhodes et al., 2005) injected into *Tg(mrc1a:GFP)* transgenic animals, examining morpholino-injected animals for *mpx* mRNA expression at 20 hpf and for FGP formation at 5 dpf. Although *mpx* is strongly reduced by *pu.1* morpholino administration (as assessed by *in situ* hybridization for endogenous *mpx*), the numbers of FGPs are not changed, suggesting primitive hematopoiesis is not required for FGPs (new Figure 5—figure supplement 1). In addition to the new figure we have added the following accompanying text:

“To test whether FGPs are derived from primitive hematopoietic progenitors, we knocked down *pu.1*, a gene required for primitive hematopoiesis (Rhodes et al., 2005), in *Tg(mrc1a:eGFP)* embryos. As shown previously, pu.1 knockdown results in loss of expression of the granulocyte-specific marker *myeloid-specific peroxidase (mpx)* above the yolk and the intermediate cell mass (n=15/15 embryos) at 20 hpf, confirming successful ablation of the primitive hematopoietic wave (Figure 5—figure supplement 1). However, FGP numbers at 5 dpf are not significantly different from controls (Figure 5—figure supplement 1), confirming that primitive hematopoiesis is not needed for FGP formation.”

b) We examined the relationship of FGPs to neural crest by imaging *Tg(sox10:dsRed); Tg(mrc1a:GFP)* double transgenic animals. The *Tg(sox10:dsRed)* transgene labels pre-migratory and migratory neural crest, as well as neural crest-derived structures, but it does not label FGPs (new Figure 4—figure supplement 2, Video 1). In addition to the new figure and Video we have added the following accompanying text to our Results section:

“Neural crest cells are an embryonic transient migratory cell population that populates various target tissues and has a multipotent capacity to differentiate into various cell types, including neurons and glia of the peripheral nervous system, craniofacial skeleton, and pigment cells (Bhatt et al., 2013; Bronner, 2012; Bronner and LeDouarin, 2012; Knight and Schilling, 2006). […] These results show that zebrafish FGPs are not neural crest derived.”

2) While macrophage origin is a major candidate, to exclude it the authors use as a "macrophage marker" the transgenic line mpx:gfp.… which, in zebrafish, only marks neutrophils and not macrophages. The only macrophage-specific transgenic reporter zebrafish lines available are based on the mpeg1 promoter (following G. Lieschke's lab) or Mfap4 promoter (Tobin lab).

In response to the reviewers comment, we examined FGPs with an mpeg1 reporter line as suggested, using a *Tg(mpeg1:Gal4-VP16); Tg(UAS:NTR-mCherry); Tg(mrc1a:eGFP)* triple-transgenic line. Although the Mpeg1 transgene marks macrophages, we do not observe any co-localization of Mpeg1 and Mrc1a in FGPs (new Figure 4). In addition to the new figure we have added the following accompanying text:

“Confocal imaging of 5 dpf or 15 dpf old *Tg(mrc1a:eGFP); Tg(mpeg1:gal4); Tg(UAS:NTR- mCherry)* triple-transgenic zebrafish shows that Mrc1a-GFP(+) FGPs and Mpeg1-mCherry(+) macrophages are distinct cell populations (Figure 4).”

3) The authors investigate definitive hematopoiesis as the source of these cells. But why ignore the primitive wave, born in the yolk sac, which produces macrophages of long-term renewal capacity (at least in mice), and could just as well be a source? Testing its involvement is straightforward, as the standard anti-Pu1 morpholino suppresses formation of this lineage.

See response #1a above. As suggested by the reviewers, we used *pu.1* morpholino knock down to suppress primitive hematopoiesis and saw no effect on FGP formation (new Figure 5—figure supplement 1).

4) Another obvious candidate source for the studied FGPs are neural crest cells. It is surprising that the authors do not even discuss this possible origin, especially since their lab addressed a similar issue in a very recently published paper (Stratman et al. Development 2017). Moreover, to prove the endothelial origin of the FGPs, they use photoconversion at 24 hpf of Kaede expressed in the Tg(Fli1:gal4; UAS:Kaede) line. Unfortunately, the fli1a gene is well known to be expressed in ectomesenchymal derivatives of NCCs (there is also primitive macrophage expression, see e.g. the yolk sac in Figure 5 of this manuscript), so that when the authors photoconvert an area medial to the eyes at 24 hpf, they necessarily photoconvert not only vascular but also neural crest cells (and possibly also primitive macrophages), which are numerous in this area at this stage.

As described in response #1b above, we used *Tg(sox10:dsRed);Tg(mrc1a:GFP)* double transgenics to show that FGPs are not derived from neural crest (new Figure 4—figure supplement 2).

In response to the reviewers comments regarding the lack of endothelial specificity of the *Tg(fli1a:gal4)* line, we repeated the photoconversion experiments using a completely endothelial- specific Tg(egfl7:gal4ff) line generated in the Kawakami lab. We photoconverted the Choroidal Vascular Plexus (CVP) in 2.5 dpf *Tg(egfl7:gal4ff); Tg(UAS:Kaede); Tg(mrc1a:eGFP)* triple- transgenic animals and, as in our experiments using the *Tg(fli1a:gal4)* driver line, observed photoconverted Kaede in FGPs at 5 dpf (new Figure 7, and Figure 7—figure supplement 2). As a control, we also repeated the photoconversion of the dorsal aorta at 24 hpf, and observed subsequent photoconverted Kaede labeling in the thymus, but not in FGPs (new Figure 5, and Figure 5—figure supplement 2). In addition to the new figures and figure panels we have added text to the Results section describing the use of the *Tg(egfl7:gal4ff)* transgenic line for dorsal aorta photoconversion and for CVP photoconversion.

5) Is there an endothelial Cre (or Cre-ER) line available that could be used to confirm the endothelial origin of these cells?

As suggested, we used a *Tg(kdrl:Cre)^s898^;Tg(-9.8actb2:LOXP-DsRed-LOXP-EGFP)^s928^*(Bertrand et al., 2010, Kikuchi et al., 2010) double transgenic “switch” line to confirm the endothelial origin of FGPs. In this double transgenic line, non-endothelial derived tissues are all dsRed+ but a small percentage of endothelial and endothelial derived structures mosaically become GFP(+) (Cre expression in the Kdrl:Cre line is not sufficient to convert all endothelium to GFP(+). As expected, a small proportion of brain vessels were GFP-positive in the double-transgenic line, but we also found that a similar small proportion of FGPs were likewise GFP-positive (new Figure 9 and Figure 9—figure supplement 1). GFP-positive FGPs were observed both in the adult zebrafish brain and also in FGPs migrating up from the CVP in 2.5 dpf animals, confirming the idea that FGPs emerge from Kdrl+ progenitors early in development. In addition to the new figure we have also added the following text to our Results section:

“The endothelial origin of FGPs was further confirmed using a *Tg(kdrl:Cre)^s898^;Tg(-9.8actb2:LOXP- DsRed-LOXP-EGFP)^s928^* (Kikuchi et al., 2010) double transgenic “switch” line (Figure 9). […] Together, these and our other results suggest that zebrafish brain FGPs have a primitive venous endothelial origin.”